# Permanent variational wave functions for bosons

J. M. Zhang,[1, 2, *] H. F. Song,[3, †] and Y. Liu[3, ‡]

[1]*Fujian Provincial Key Laboratory of Quantum Manipulation and New Energy Materials,*
*College of Physics and Energy, Fujian Normal University, Fuzhou 350007, China*
[2]*Fujian Provincial Collaborative Innovation Center for Optoelectronic*
*Semiconductors and Efficient Devices, Xiamen, 361005, China*
[3]*Laboratory of Computational Physics, Institute of Applied Physics and Computational Mathematics, Beijing 100088, China*

We study the usefulness of the permanent state as variational wave functions for bosons, which is the bosonic counterpart of the Slater determinant state for fermions. For a system of $N$ identical bosons, a permanent state is constructed by taking a set of $N$ arbitrary (not necessarily orthonormal) single-particle orbitals, forming their product and then symmetrizing it. It is found that for the one-dimensional Bose-Hubbard model with the periodic boundary condition and at unit filling, the exact ground state can be very well approximated by a permanent state, in that the permanent state has high overlap (at least 0.96 for 12 particles and 12 sites) with the exact ground state and can reproduce both the ground state energy and the single-particle correlators to high precision. For more general models, we have devised an optimization algorithm to find the optimal set of single-particle orbitals to minimize the variational energy or maximize the overlap with a given state. It turns out that quite often the ground state of a bosonic system can be well approximated by a permanent state by all the criterions of energy, overlap, and correlation functions. And even if the error is apparent, it can often be remedied by including more configurations, i.e., by allowing the variational wave function to be the superposition of multiple permanent states. All these suggest that permanent states are very effective as variational wave functions for many bosonic systems.

## I. INTRODUCTION

The Hartree-Fock approximation for fermions is a paradigm in quantum mechanics [1–6]. Conceptually, it is very simple. It is a variational method. For an $N$-fermion system, one just takes $N$ orthonormal single-particle orbitals $\{\phi_i | 1 \leq i \leq N\}$, constructs the product state $\phi_1(x_1)\phi_2(x_2)\ldots\phi_N(x_N)$, and then anti-symmetrizes it to obtain the Slater determinant wave function

$$
\begin{aligned}
&\Xi(x_1, x_2, \ldots, x_N) \\
&= \frac{1}{\sqrt{N!}} \sum_{P \in S_N} (-1)^P \phi_{P_1}(x_1)\phi_{P_2}(x_2)\ldots\phi_{P_N}(x_N) \\
&= \frac{1}{\sqrt{N!}} \det(\phi_i(x_j)).
\end{aligned}
\tag{1}
$$

Here $S_N$ denotes the symmetric group of degree $N$. By construction, the determinant state satisfies the anti-symmetry condition and constitutes a legitimate wave function for a collection of identical fermions. With the variational wave function built in this way, the rest work is an optimization problem. One has to choose the $N$ orthonormal orbitals optimally so as to minimize the energy expectation value of the $N$-body variational state.

It is a natural idea to generalize this approach to bosons. One can take $N$ single-particle orbitals $\{\phi_i | 1 \leq i \leq N\}$, form their product, but then symmetrize it to

obtain the following state,

$$
\begin{aligned}
&\Phi(x_1, x_2, \ldots, x_N) \\
&= \frac{1}{\sqrt{N!}} \sum_{P \in S_N} \phi_{P_1}(x_1)\phi_{P_2}(x_2)\ldots\phi_{P_N}(x_N) \\
&= \frac{1}{\sqrt{N!}} \mathrm{per}(\phi_i(x_j)),
\end{aligned}
\tag{2}
$$

which we shall refer to as a permanent state. Unlike the fermionic case, here because of the symmetry instead of anti-symmetry condition, the single-particle orbitals are not necessarily orthogonal to each other [7], but could even be identical. In the extremal case in which all the orbitals are constrained to be the same, we have the Gross-Pitaevskii approximation [8, 9], which has been proven to be very successful for weakly interacting bose gases [10, 11]. However, for more general systems, such as the Bose-Hubbard model which we shall study below, the Gross-Pitaevskii approximation is too restrictive and we had better allow more freedom for the $N$ orbitals.

The idea seems very simple. However, probably because mathematically the permanent of a matrix lacks many of the nice properties of the determinant, such an approach has rarely been put into practice. As far as we know, the very limited literature starts with two papers of Romanovsky *et al.* in 2004 and 2006 [12, 13]. They employed the permanent state as variational wave functions for some few-boson systems in two-dimensional harmonic traps. They went beyond the Gross-Pitaevskii approximation by allowing each particle to occupy a different orbital, for which they coined the term unrestricted Bose-Hartree-Fock approximation. However, because of the perceived high complexity of the self-consistency equations, they did not seek self-consistent orbitals, but

———————

* wdlang06@163.com
† song_haifeng@iapcm.ac.cn
‡ liu_yu@iapcm.ac.cn

prescribed them as displaced Gaussians. Subsequently, the self-consistency equations for the orbitals were derived by Heimsoth [14, 15]. Unfortunately, the formalism was still unnecessarily complicated, and he did not even implemented the Ryser algorithm for permanent computation. Consequently, he could handle at most six particles.

In this paper, we resume research in this vein. We shall see that permanent states are very effective as variational wave functions. Moreover, the difficulties above can be evaded if the problem is formulated appropriately. In particular, it is not that time consuming to deal with permanents. With a typical laptop computer, we can handle up to 12 particles in reasonable time.

This paper is organized as follows. First in Sec. II, we review the connection between the first and second quantization formalisms, and we shall establish some analytic facts about the permanent state. Then in Sec. III, we show that for the one-dimensional Bose-Hubbard model with periodic boundary condition and at unit filling, which is the standard setting for studying the superfluid-Mott insulator transition, the permanent state can be a very good approximation of the exact ground state. It is good not only by the usual energy criterion, but also by the more stringent criterions of overlap and correlation functions. For a Bose-Hubbard model with 12 particles on 12 sites, the energy-minimizing permanent state with prescribed orbitals has an overlap with the exact ground state as large as 0.96 in the worst case. Of course one should not be satisfied with prescribed orbitals. It is desirable to have more flexibility and presumably the numbers could be further improved if the orbitals are really unrestricted. We thus propose an iteration algorithm in Sec. IV for searching for the optimal set of orbitals minimizing the energy. The equations are equivalent to what Heimsoth derived [14, 15]. However, because of the different point of view, our derivation is more elementary and straightforward, and the formulation is more amenable for numerical implementation. Moreover, our formalism can treat the multi-configuration case, i.e., the case when the variational wave function is a linear combination of multiple permanent states, easily. The algorithm can actually be employed to solve another optimization problem, namely, for a given wave function, finding the single- or multi-configurational variational wave function most close to it, i.e., having the largest possible overlap with it. Although this problem is rarely studied in the literature and is not our focus in this paper, it should be a meaningful question for studying the structure of a bosonic wave function. With the optimization algorithm, we can tackle more general models. This is what we do in Sec. V. We shall see that in many cases, a single- or multi-configurational variational wave function is a very good approximation of the exact ground state of the system. Finally, we conclude in Sec. VI with some open problems.

## II. PERMANENT WAVE FUNCTIONS

For the sake of simplicity, let us assume a finite-dimensional single-particle Hilbert space

$$\mathcal{H} = \text{span}\{|x\rangle, 1 \leq x \leq L\}, \tag{3}$$

where $|x\rangle$ are orthonormal basis vectors. The associated creation (annihilation) operators will be denoted as $a_x^\dagger$ ($a_x$). They satisfy the usual commutation relations. A generic (not necessarily normalized) single-particle state or a single-particle orbital in this space is $|\phi\rangle = \sum_{x=1}^L |x\rangle\langle x|\phi\rangle = \sum_{x=1}^L \phi(x)|x\rangle$. The associated creation operator is $a_\phi^\dagger = \sum_{x=1}^L \phi(x)a_x^\dagger$.

For an $N$-boson system, the many-body Hilbert space is spanned by the orthonormal Fock states

$$|\mathbf{n}\rangle = \frac{(a_1^\dagger)^{n_1}(a_2^\dagger)^{n_2}\ldots(a_L^\dagger)^{n_L}}{\sqrt{n_1!n_2!\ldots n_L!}}|vac\rangle. \tag{4}$$

where $\mathbf{n} \equiv (n_1, n_2, \ldots, n_L)$ is an $L$-tuple with $n_x \geq 0$ and $\sum_{x=1}^L n_x = N$. The number of such Fock states or the dimension of the many-body Hilbert space is

$$\mathcal{D} = \binom{N+L-1}{N} = \frac{(N+L-1)!}{N!(L-1)!}. \tag{5}$$

A generic $N$-boson state $|\Psi\rangle$ expands as $|\Psi\rangle = \sum_{\mathbf{n}} C(\mathbf{n})|\mathbf{n}\rangle$. In first quantization, the same state is expressed by the wave function $\Psi(x_1, x_2, \ldots, x_N)$, with $1 \leq x_i \leq L$. Under the action of particle permutations, the coordinate tuples $\mathbf{x} \equiv (x_1, x_2, \ldots, x_N)$ break into different equivalent classes labelled by the occupation tuple $\mathbf{n}$. The wave function should be constant on each class. We say a coordinate tuple $\mathbf{x}$ belongs to $\mathbf{n}$ and denote it as $\mathbf{x} \vdash \mathbf{n}$ if in $\mathbf{x}$, the value $x$ appears $n_x$ times. The cardinality of the class $\mathbf{n}$ is $N!/\prod_{x=1}^L n_x!$, and thus by considering the norm of $\Psi$ in both the first and second quantization form, we have [16]

$$\Psi(\mathbf{x}) = \sqrt{\frac{n_1!n_2!\ldots n_L!}{N!}}C(\mathbf{n}), \quad \mathbf{x} \in \mathbf{n}. \tag{6}$$

With this formula, one can convert a wave function in the first quantization form to the second quantization form, and vice versa.

Now suppose we have a set of $N$ arbitrary orbitals $\{\phi_i(x), 1 \leq i \leq N\}$. The simplest symmetric $N$-particle wave function one can construct out of them is the permanent state in (2). We shall use the notation

$$\Phi = \hat{\mathcal{S}}(\phi_1, \phi_2, \ldots, \phi_N) \tag{7}$$

to indicate that $\Phi$ is built out of the orbitals $\{\phi_i(x), 1 \leq i \leq N\}$ according to the product and symmetrization procedure in (2). By the correspondence (6), it is easy to show that in second quantization, this state has the expression

$$|\Phi\rangle = \prod_{i=1}^N a_{\phi_i}^\dagger |vac\rangle. \tag{8}$$

This form should reminds us of the standard Fock states. They are also permanent states, but with orthonormal orbitals.

Here we emphasize that in this paper *we abandon the orthogonality and normalization of the orbitals.* This is a fundamental difference between the current approach and the multiconfigurational Hartree theory for bosons (M-CHB) [17], which although also treats the single-particle orbitals as variational parameters, insists on their orthogonality and normalization. One apparent advantage of using non-orthogonal orbitals is that the expression of the wave function is more *compact* [18]—A single permanent state in the form of (8) would expand into a multitude of Fock states if the orbitals $\phi_i$ are expanded in terms of an orthonormal basis. The downside is that we have to compute the permanents of overlap matrices, which is expensive in CPU time.

### A. Five simple propositions

At least five simple facts about a permanent state can be easily established. Although they are not much used in this paper, we collect them here for completeness.

The first one seems trivial or only of academic interest. Intuitively, for a given set of nonzero orbitals $\{\phi_i, 1 \leq i \leq N\}$, the permanent state constructed according to (2) or (8) is nonzero too, or, there is no danger of perfect cancellation between the terms in the sum of (2). This is indeed the case. We have

**Proposition 1.** *The $N$-particle permanent state $\Phi$ constructed with $N$ nonzero orbitals $\{\phi_{1 \leq i \leq N}\}$ according to (2) is necessarily non-vanishing.*

*Proof.* Consider the inner product of $\Phi$ with the condensate-type symmetric state

$$\Omega = v(x_1)v(x_2)\ldots v(x_N), \tag{9}$$

where $v \in \mathcal{H}$ is an arbitrary single-particle state. We have

$$\frac{1}{\sqrt{N!}}\langle\Phi|\Omega\rangle = \prod_{i=1}^{N}\langle\phi_i|v\rangle. \tag{10}$$

The product on the right hand vanishes if and only if $v$ is orthogonal to some $\phi_i$, or $v \in \bigcup_{i=1}^{N}\ker(\phi_i)$. Here by an abuse of notation, we have also used $\phi_i$ to denote the linear functional $\langle\phi_i|\cdot\rangle$. The kernel of this functional $\ker(\phi_i)$ is simply the hyperplane orthogonal to $\phi_i$. By the well-known folklore in mathematics that a vector space over $\mathbb{C}$ cannot be the union of a finite number of proper subspaces [19], we know the union $\bigcup_{i=1}^{N}\ker(\phi_i)$ cannot cover the whole space $\mathcal{H}$, and for some $v$ the product is non-vanishing, which in turn means that $\Phi$ must be non-vanishing. $\square$

The second one is about uniqueness. Given a set of orbitals $\{\phi_i, 1 \leq i \leq N\}$, one can construct a permanent

state according to (2) or (8). One might ask whether the same permanent state can be built with a different set of orbitals. Here of course, two sets of orbitals should be deemed equivalent if they differ just by a permutation or some linear scaling. The answer is no as we have

**Proposition 2.** *Suppose a non-vanishing $N$-particle permanent state $\Phi$ can be constructed with two sets of orbitals $\{f_{1 \leq i \leq N}\}$ and $\{g_{1 \leq i \leq N}\}$, i.e.,*

$$\Phi = \hat{\mathcal{S}}(f_1, f_2, \ldots, f_N) = \hat{\mathcal{S}}(g_1, g_2, \ldots, g_N), \tag{11}$$

*then for some permutation $\sigma \in S_N$, $f_i \propto g_{\sigma(i)}$ for all $1 \leq i \leq N$.*

*Proof.* We provide two proofs. The first one is elementary but lengthy. Again, let us consider the inner product of $\Phi$ with the condensate-type state in (9). We have

$$\frac{1}{\sqrt{N!}}\langle\Phi|\Omega\rangle = \prod_{i=1}^{N}\langle f_i|v\rangle = \prod_{i=1}^{N}\langle g_i|v\rangle. \tag{12}$$

Now suppose $v$ is orthogonal to $f_1$, i.e., $\langle f_1|v\rangle = 0$, or $v \in \ker(f_1)$. The equality above implies $v \in \bigcup_{i=1}^{N}\ker(g_i)$. As this holds for any $v \in \ker(f_1)$, we obtain $\ker(f_1) \subseteq \bigcup_{i=1}^{N}\ker(g_i)$, which in turn means

$$\ker(f_1) = \ker(f_1)\bigcap\left(\bigcup_{i=1}^{N}\ker(g_i)\right)$$
$$= \bigcup_{i=1}^{N}\left(\ker(f_1)\bigcap\ker(g_i)\right). \tag{13}$$

Again by the well-known folklore that a vector space cannot be the union of a finite number of proper subspaces [19], we know there must be some $i$ such that $\ker(f_1) = \ker(f_1)\bigcap\ker(g_i)$, or $\ker(f_1) = \ker(g_i)$, which means $f_1 \propto g_i$. By permutation or relabeling, we can assume $i = 1$ and simply $f_1 = g_1$.

We then wish to factor out $f_1$ and $g_1$ in (12) to get

$$\prod_{i=2}^{N}\langle f_i|v\rangle = \prod_{i=2}^{N}\langle g_i|v\rangle \tag{14}$$

for all $v \in \mathcal{H}$. This already holds for $v \notin \ker(f_1)$. To show that it actually holds also for $v \in \ker(f_1)$, consider $v \in \ker(f_1)$ and $w \notin \ker(f_1)$. For any nonzero $t \in \mathbb{C}$, $v + tw \notin \ker(f_1)$, otherwise $w = [(v + tw) - v]/t$ would be in $\ker(f_1)$. By (14), we have then

$$\prod_{i=2}^{N}\langle f_i|v + tw\rangle = \prod_{i=2}^{N}\langle g_i|v + tw\rangle \tag{15}$$

for all $t \neq 0$. That is, two polynomials of $t$ evaluate to the same value for all $t \neq 0$. This means that the two polynomials are actually the same. In particular, their constant terms are identical. We have thus proven that (14) holds for all $v \in \ker(f_1)$ too. The proposition is then proven by induction.

The second proof is more direct. Let $|f_i\rangle = \sum_{j=1}^{L} f_{ij}|j\rangle$ and similarly $|g_i\rangle = \sum_{j=1}^{L} g_{ij}|j\rangle$. By (8) and (11),

$$\Phi = \prod_{i=1}^{N} \left( \sum_{j=1}^{L} f_{ij} a_j^\dagger \right) |vac\rangle = \prod_{i=1}^{N} \left( \sum_{j=1}^{L} g_{ij} a_j^\dagger \right) |vac\rangle. \quad (16)$$

The fact that the $a_j^\dagger$ operators commute and the Fock basis states are linearly independent implies the polynomial equality

$$\prod_{i=1}^{N} \left( \sum_{j=1}^{L} f_{ij} z_j \right) = \prod_{i=1}^{N} \left( \sum_{j=1}^{L} g_{ij} z_j \right), \quad (17)$$

where $z_j$ are indeterminates. The proposition is proven by the unique factorization theorem of multivariate polynomials over $\mathbb{C}$ [20]. $\qquad \square$

To appreciate Proposition 2, one should note that for fermions, a Slater determinant state constructed out of $N$ orbitals is determined by the subspace (a point on the so-called Grassmannian manifold [21]) spanned by the orbitals, not by the orbitals themselves. The orbitals are just a basis of the subspace. Another basis would yield the same Slater determinant state up to a global constant.

By Proposition 2, the permanent state $\Phi$ is invariant under the permutation $f_i \to f_{\sigma_i}$ with $\sigma$ being an arbitrary permutation, and the scaling $f_i \to \lambda_i f_i$ with the scaling factors satisfying the condition $\prod_{i=1}^{N} \lambda_i = 1$. This allows us to count the degrees of freedom [22] of a permanent state as

$$d = NL - (N - 1) = N(L - 1) + 1. \quad (18)$$

Apparently, like not every $N$-fermion state is a Slater determinant state, not every $N$-boson state is a permanent state. However, in the special case of $\dim \mathcal{H} = L = 2$, this is indeed the case. We have

**Proposition 3.** *Suppose the single-particle Hilbert space is of dimension 2, i.e., $\dim \mathcal{H} = L = 2$, then every $N$-boson state is a permanent state.*

*Proof.* In this case, the $N$-boson Hilbert space is of dimension $N + 1$ and a basis is the Fock states $\{(a_1^\dagger)^i (a_2^\dagger)^{N-i}|vac\rangle, 0 \le i \le N\}$. But by (18), the number of degrees of freedom of a permanent state is also $N + 1$, so the proposition is anticipated by dimension counting. To prove it rigorously, we expand an arbitrary state $\Psi$ as

$$\Psi = (a_2^\dagger)^m \sum_{k=0}^{N-m} c_k (a_1^\dagger)^{N-m-k} (a_2^\dagger)^k |vac\rangle. \quad (19)$$

Here $m$ is an integer between 0 and $N$, and $c_0 \ne 0$. The fact that $a_1^\dagger$ and $a_2^\dagger$ commute motivates us to consider the polynomial $P(z) = \sum_{k=0}^{N-m} c_k z^{N-m-k}$. Let it factorize as

$$P(z) = c_0 \prod_{j=1}^{N-m} (z - z_j). \quad (20)$$

Then it is easy to see that $\Psi$ factorize as

$$\Psi = c_0 (a_2^\dagger)^m \prod_{j=1}^{N-m} (a_1^\dagger - z_j a_2^\dagger)|vac\rangle. \quad (21)$$

By (8), we see it is a permanent state and can read off the single-particle orbitals. $\qquad \square$

Proposition 3 means that for a two-site Bose-Hubbard model [23], which is a canonical model for studying the Bose-Josephson effect, any state is a permanent state. The proposition also reminds us of a similar proposition for fermions [24, 25]. Although an arbitrary fermionic wave function is not generally a Slater determinant state, for the case of $N$ fermions in $L = N+1$ orbitals, the wave function is necessarily a Slater determinant.

Proposition 3 is about the case when the dimension of the single-particle Hilbert space is minimal (but still nontrivial). Similarly, when the particle number is minimal (but still nontrivial), i.e., when $N = 2$, we have

**Proposition 4.** *Suppose $N = 2$ and $\dim \mathcal{H} = L$, then every $N$-boson state can be written as the sum of at most $\lfloor (L+1)/2 \rfloor$ permanent states. Here $\lfloor x \rfloor$ is the floor function denoting the greatest integer less than or equal to $x$.*

*Proof.* For $N = 2$, the bosonic wave function $\Psi(x_1, x_2)$ with $1 \le x_i \le L$ can be considered as a complex symmetric matrix. By the Autonne-Takagi theorem [26], it can be factorized as

$$\Psi(x_1, x_2) = \sum_{j=1}^{L} \sqrt{D_j} f_j(x_1) f_j(x_2), \quad (22)$$

where $f_j$ are a set of orthonormal functions and $D_j$ are a set of non-negative numbers in decreasing order. The functions $f_j$ are actually the so-called natural orbitals, i.e., eigenvectors of the one-body reduced density matrix $\rho = 2\Psi\Psi^\dagger$ associated with $\Psi$, and $2D_j$ are the occupation numbers. In the following, we shall assume that $\Psi$ is normalized, i.e., $\langle \Psi|\Psi \rangle = 1$, so that $\sum_{j=1}^{L} D_j = 1$.

In the form of (22), the wave function $\Psi$ is already a sum of $L$ permanent states. Now for any pair $j \ne k$, we can combine $\sqrt{D_j} f_j(x_1) f_j(x_2)$ and $\sqrt{D_k} f_k(x_1) f_k(x_2)$ into a single permanent state, i.e.,

$$\hat{S}(u, v) = \frac{1}{\sqrt{2}} [u(x_1)v(x_2) + v(x_1)u(x_2)]$$

with (note that by Proposition 2, this is essentially the only solution)

$$u \equiv \frac{1}{\sqrt[4]{2}} (\sqrt[4]{D_j} f_j + i \sqrt[4]{D_k} f_k),$$

$$v \equiv \frac{1}{\sqrt[4]{2}} (\sqrt[4]{D_j} f_j - i \sqrt[4]{D_k} f_k).$$

The proposition is then proven by noting that in (22), when $L$ is even, we have $L/2$ pairs and when $L$ is odd, we have $(L-1)/2$ pairs and an extra unpaired term, which is already a permanent state. $\qquad \square$

It is easy to see that the number $\lfloor (L + 1)/2 \rfloor$ in Proposition 4 is not only sufficient but also necessary for a generic 2-boson state. For a generic state, the matrix $\Psi$ is nonsingular or full-ranked. On the other hand, a permanent state is at most 2-ranked. Hence, by the subadditivity property of the rank of a matrix $[\text{rank}(A + B) \leq \text{rank}(A) + \text{rank}(B)]$, we need at least $\lfloor (L+1)/2 \rfloor$ permanent states to fully recover the original state. However, in practice, one might just need a sufficiently good approximation of the original state. The question is then, by taking the sum of $M < \lfloor (L+1)/2 \rfloor$ permanent states, i.e., by constructing a state in the form of

$$\Omega = \sum_{\alpha=1}^{M} \hat{\mathcal{S}}(u^{(\alpha)}, v^{(\alpha)}), \qquad (23)$$

where $u^{(\alpha)}$ and $v^{(\alpha)}$ are arbitrary orbitals, to what extent can we approximate a target function $\Psi$? Quantitatively, what is the maximal value of the overlap

$$O = \frac{|\langle \Omega | \Psi \rangle|^2}{\langle \Omega | \Omega \rangle \langle \Psi | \Psi \rangle} \qquad (24)$$

achievable with such an $M$-configuration state? For this problem, we have

**Proposition 5.** *Suppose $N = 2$, $\dim \mathcal{H} = L$, and $M \leq \lfloor (L + 1)/2 \rfloor$. For the target state $\Psi$ in (22), the largest possible value of the overlap of an $M$-configuration state in the form of (23) with it is*

$$O_{max} = \max_{\Omega} \frac{|\langle \Omega | \Psi \rangle|^2}{\langle \Omega | \Omega \rangle \langle \Psi | \Psi \rangle} = \sum_{j=1}^{2M} D_j. \qquad (25)$$

*Proof.* Like $\Psi$ in (22), the $M$-configuration state $\Omega$ in (23) is also symmetric and can also be factorized as

$$\Omega = \sum_{j=1}^{2M} \sqrt{C_j} \varphi_j(x_1) \varphi_j(x_2), \qquad (26)$$

where $\varphi_{1 \leq j \leq 2M}$ are a set of orthonormal vectors which can be extended into a complete orthonormal basis $\varphi_{1 \leq j \leq L}$, and $C_j \geq 0$ are ordered in decreasing order. Without loss of generality, let us assume $\Omega$ is normalized so that $\sum_{j=1}^{2M} C_j = 1$. Note that by construction $\Omega$ is at most of rank $2M$, and thus here in (26) we have at most $2M$ nonzero terms. For the overlap between $\Omega$ and $\Psi$, we have

$$O = \left| \sum_{j=1}^{2M} \sqrt{C_j} \langle \varphi_j \varphi_j | \Psi \rangle \right|^2$$

$$\leq \left( \sum_{j=1}^{2M} C_j \right) \left( \sum_{j=1}^{2M} |\langle \varphi_j \varphi_j | \Psi \rangle|^2 \right)$$

$$= \sum_{j=1}^{2M} |\langle \varphi_j \varphi_j | \Psi \rangle|^2 \leq \sum_{j=1}^{2M} \sum_{k=1}^{L} |\langle \varphi_j \varphi_k | \Psi \rangle|^2 = \text{Tr}(AA^{\dagger}),$$

where $A \equiv V^{\dagger} \Psi W^*$, with $V$ the $L \times 2M$ matrix whose columns are $\varphi_{1 \leq j \leq 2M}$ and $W$ the $L \times L$ matrix whose columns are $\varphi_{1 \leq k \leq L}$. Note that $W$ is unitary. We have

$$\text{Tr}(AA^{\dagger}) = \text{Tr}(V^{\dagger} \Psi W^* W^T \Psi^{\dagger} V)$$

$$= \text{Tr}(V^{\dagger} \Psi \Psi^{\dagger} V) \leq \sum_{j=1}^{2M} D_j. \qquad (27)$$

Here we used the Ky-Fan inequality [27] for the matrix $\Psi \Psi^{\dagger}$, which states the for an $L \times L$ hermitian matrix, the sum of its $m$ diagonal elements is less than or equal to the sum of its $m$ largest eigenvalues, for all $1 \leq m \leq L$. We have thus proven that the overlap is upper bounded by $\sum_{j=1}^{2M} D_j$. That this upper bound can be achieved is obvious, as we can just take the first $2M$ terms in (22) and by Proposition 4 it is an $M$-configuration state. $\square$

In general, the occupation numbers $D_j$ decrease fast and a truncation of (22) with a very limited number of terms can yield a good approximation of the original state.

## B. Basic formulae

A generic many-body Hamiltonian is of the form

$$H = H_1 + H_2 = \sum_{i=1}^{N} K(i) + \sum_{1 \leq i < j \leq N} U(i, j). \quad (28)$$

Here the first sum is over each particle, with $K(i)$ denoting the sum of the kinetic energy and the external potential of the $i$th particle, while the second sum is over each pair, with $U(i, j)$ denoting the interaction between the $i$th and $j$th particle. The quantity of primary interest is the expectation value of $H$ with respect to a permanent state (2), i.e.,

$$E_{var} = \frac{\langle \Phi | H | \Phi \rangle}{\langle \Phi | \Phi \rangle} = \frac{\langle \Phi | H_1 | \Phi \rangle + \langle \Phi | H_2 | \Phi \rangle}{\langle \Phi | \Phi \rangle}. \quad (29)$$

It is straightforward to calculate the denominator and the numerator here. But for the sake of generality and in view of the extension to the multiconfigurational case below, let us consider the off-diagonal matrix elements instead of the diagonal matrix elements. Suppose $\Phi^{(1)}$ and $\Phi^{(2)}$ are two permanent states constructed with two sets of orbitals $\{\phi_{1 \leq i \leq N}^{(\alpha)}, \alpha = 1, 2\}$, i.e.,

$$\Phi^{(\alpha)} = \hat{\mathcal{S}}(\phi_1^{(\alpha)}, \phi_2^{(\alpha)}, \ldots, \phi_N^{(\alpha)}), \quad \alpha = 1, 2. \quad (30)$$

Let us first define the $N \times N$ overlap matrix of the orbitals,

$$A_{ij} = \langle \phi_i^{(1)} | \phi_j^{(2)} \rangle, \quad 1 \leq i, j \leq N. \qquad (31)$$

We have then

$$
\begin{aligned}
&\langle \Phi^{(1)}|\Phi^{(2)}\rangle \\
&= \frac{1}{N!} \sum_{P,Q\in S_N} \langle \phi_{P_1}^{(1)}|\phi_{Q_1}^{(2)}\rangle \langle \phi_{P_2}^{(1)}|\phi_{Q_2}^{(2)}\rangle \dots \langle \phi_{P_N}^{(1)}|\phi_{Q_N}^{(2)}\rangle \\
&= \sum_{R\in S_N} \langle \phi_1^{(1)}|\phi_{R_1}^{(2)}\rangle \langle \phi_2^{(1)}|\phi_{R_2}^{(2)}\rangle \dots \langle \phi_N^{(1)}|\phi_{R_N}^{(2)}\rangle \\
&= \mathrm{per}(A).
\end{aligned} \tag{32}
$$

Here $\mathrm{per}(A)$ denotes the permanent of the matrix $A$. As for the matrix elements of $H_1$,

$$
\begin{aligned}
\langle \Phi^{(1)}|H_1|\Phi^{(2)}\rangle &= \frac{N}{N!} \sum_{P,Q\in S_N} \langle \phi_{P_1}^{(1)}|K|\phi_{Q_1}^{(2)}\rangle \prod_{i=2}^{N} \langle \phi_{P_i}^{(1)}|\phi_{Q_i}^{(2)}\rangle \\
&= \sum_{i_1=1}^{N}\sum_{j_1=1}^{N} \langle \phi_{i_1}^{(1)}|K|\phi_{j_1}^{(2)}\rangle \, \mathrm{per}(A; i_1|j_1), 
\end{aligned} \tag{33}
$$

where $\mathrm{per}(A; i_1|j_1)$ denotes the permanent of the $(N-1)\times(N-1)$ minor obtained by deleting the $i_1$th row and the $j_1$th column from $A$. Similarly, for the matrix elements of $H_2$,

$$
\begin{aligned}
&\langle \Phi^{(1)}|H_2|\Phi^{(2)}\rangle \\
&= \frac{N(N-1)}{2(N!)} \sum_{P,Q\in S_N} \langle \phi_{P_1}^{(1)}\phi_{P_2}^{(1)}|U|\phi_{Q_1}^{(2)}\phi_{Q_2}^{(2)}\rangle \prod_{i=3}^{N} \langle \phi_{P_i}^{(1)}|\phi_{Q_i}^{(2)}\rangle \\
&= \frac{1}{2}\sum_{i_1\neq i_2, 1}^{N}\sum_{j_1\neq j_2,1}^{N} \langle \phi_{i_1}^{(1)}\phi_{i_2}^{(1)}|U|\phi_{j_1}^{(2)}\phi_{j_2}^{(2)}\rangle \\
&\qquad\qquad\qquad \times \mathrm{per}(A; i_1,i_2|j_1,j_2).
\end{aligned} \tag{34}
$$

Here in the last line, the summation $\sum_{i_1\neq i_2,1}^{N}$ means that $i_1$ and $i_2$ both run from 1 to $N$, but they must take different values. The second summation is interpreted similarly. By $\mathrm{per}(A; i_1,i_2|j_1,j_2)$ we mean the permanent of the $(N-2)\times(N-2)$ minor of $A$ obtained by deleting row $i_1$, $i_2$ and column $j_1$, $j_2$.

Above we see that to calculate the norm and physical expectation values of a permanent state, we have to calculate the permanent of the overlap matrix $A$ and those of its minors. This is the price we have to pay for working with non-orthonormal orbitals.

As is generally believed, unlike the determinant of a matrix, the permanent of the matrix cannot be calculated in polynomial time. Currently, the best known general exact algorithm is the Ryser algorithm [28], which reduces the naive $N \cdot N!$ evaluations to $O(N^2 2^{N-1})$. By using the Gray code, a further reduction by a factor of $N$ can be achieved [29]. This is the algorithm we use in this work [30]. With this algorithm, it takes about 1 sec (0.01 sec) to calculate the permanent of a $22\times22$ ($15\times15$, respectively) real-valued matrix on a commercial laptop computer and with MATLAB. We mention that in this paper, all calculation is done with MATLAB but without invoking the parallel computing toolbox.

## III. BOSE-HUBBARD MODEL AT UNIT FILLING

To see whether a permanent wave function can be a good approximation of the ground state of a bosonic system, we take the one-dimensional Bose-Hubbard model with the periodic boundary condition and at unit filling as a case study. The Hamiltonian, as is often written in the second-quantization formalism, reads

$$
H = -\sum_{x=0}^{L-1}(a_x^\dagger a_{x+1} + a_{x+1}^\dagger a_x) + \frac{g}{2}\sum_{x=0}^{L-1} a_x^\dagger a_x^\dagger a_x a_x. \tag{35}
$$

Here the single-particle Hilbert space $\mathcal{H}$ is spanned by the orthonormal site (Wannier) states $\{|x\rangle, 0 \le x \le L-1\}$. The periodic boundary condition means $|x\rangle = |x+L\rangle$. Note that we have taken the hopping strength as the unit of energy, and the Hamiltonian depends only on the parameter $g \ge 0$, which characterizes the on-site interaction strength.

For our purpose, it is often more convenient to work with the first-quantization formalism of (28). The corresponding $K$ and $U$ operators have matrix elements as

$$
K_{x,x'} = -(\delta_{x,x'+1} + \delta_{x,x'-1}), \tag{36}
$$
$$
U_{x_1 x_2, x_1' x_2'} = g\delta_{x_1,x_2}\delta_{x_1,x_1'}\delta_{x_1,x_2'}. \tag{37}
$$

In this section, we shall confine ourself to the unit filling case, namely, the case when the particle number $N = L$. The commensurate condition and the translation symmetry suggest putting the $i$th particle in an orbital centered at site $i$, with all the orbitals of the same shape and related to each other by translations. Specifically,

$$
\phi_i(x) = \phi(x-i), \qquad 1 \le i \le N, \tag{38}
$$

for some function $\phi$. Here the periodic boundary condition requires $\phi(x) = \phi(x+L)$. The total variational wave function (VWF) is then determined by the single-particle orbital $\phi$. To construct an $L$-periodic function $\phi$, we can choose an arbitrary primitive function $\chi(x)$ defined on the whole axis $-\infty < x < +\infty$, and form the superposition

$$
\phi(x) = \sum_{j=-\infty}^{\infty} \chi(x-jL). \tag{39}
$$

We have tried two types of primitive functions, i.e., the Lorentz function and the exponential function,

$$
\chi^{(l)}(x;\lambda) = (1+\lambda^2 x^2)^{-1}, \tag{40a}
$$
$$
\chi^{(e)}(x;\lambda) = e^{-\lambda|x|}, \tag{40b}
$$

where $\lambda \ge 0$ is a parameter controlling the width of the functions. For these two simple types of primitive functions, the summation in (39) can be carried out analytically and yields

$$
\phi^{(l)}(x;\lambda) = \frac{1-e^{-4\pi/L\lambda}}{|1-e^{-2\pi/L\lambda}e^{i2\pi x/L}|^2}, \tag{41a}
$$

$$
\phi^{(e)}(x;\lambda) = \frac{e^{-\lambda x}+e^{-\lambda(L-x)}}{1-e^{-\lambda L}}, \quad 0 \le x \le L. \tag{41b}
$$

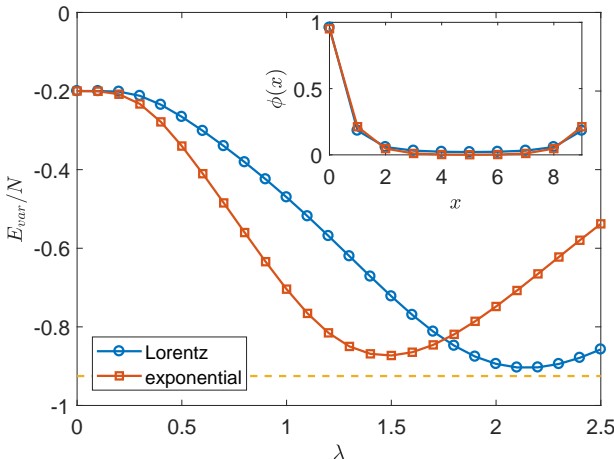

FIG. 1. (Color online) Energy expectation value (per particle) of the permanent variational state constructed with either the Lorentz-type (40a) or the exponential-type (40b) primitive orbitals. The horizontal dashed line indicates the exact ground state energy calculated by exact diagonalization. The inset shows the optimal orbitals corresponding to the minima of the curves $E_{var}(\lambda)$. The parameters are $N = L = 10$ and $g = 4$.

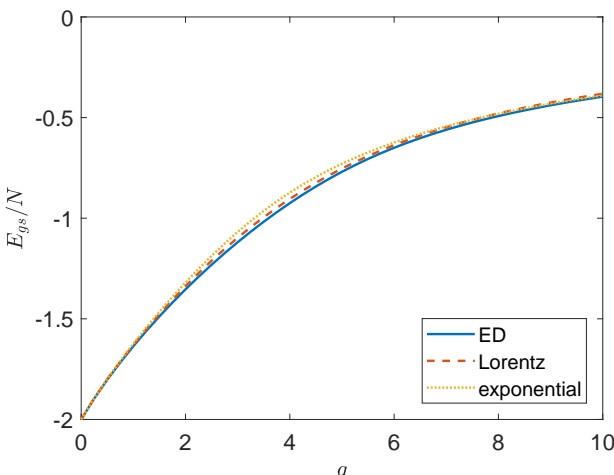

FIG. 2. (Color online) Ground state energy (per particle) estimated with the permanent variational state constructed with either the Lorentz-type (40a) or the exponential-type (40b) primitive orbitals. The solid line is the exact value obtained by exact diagonalization (ED). In the region of $g \lesssim 8$, the Lorenztian-type orbital yields a better estimate; while in the region of $g \gtrsim 8$, the exponential-type orbital is better, although this latter fact is hardly visible in the figure. The parameters are $N = L = 10$. The dimension of the Hilbert space is $92\,378$.

Note that in the limit of $\lambda \to 0$, both $\phi^{(l)}$ and $\phi^{(e)}$ reduce to the zero-momentum Bloch state on the periodic lattice, while in the opposite limit of $\lambda \to \infty$, both of them reduce to the Kronecker delta function $\delta_{x,0}$.

The strategy is then simply to vary the parameter $\lambda$ and calculate the variational energy $E_{var}(\lambda; g)$ as a func-

tion of $\lambda$ by using the formulae in Sec. II B. Here and henceforth, the dependence of $E_{var}$ on the primitive orbitals should be understood tacitly. By the variational principle, $E_{var}$ is always above the exact ground state energy $E_{gs}^{ext}$. The concern is whether the minimum of $E_{var}$ can be sufficiently close to $E_{gs}^{ext}$. A case study with $N = L = 10$ and $g = 4$ is shown in Fig. 1. We see that for both types of primitive orbitals, at some value of $\lambda$, $E_{var}$ dips towards the horizontal line indicating $E_{gs}^{ext}$. At the minima, the relative error is 2% and 4%, respectively, for the Lorentz-type and exponential-type orbital. This is very encouraging and turns out to be typical.

By determining the minimum of the curve $E_{var}(\lambda; g)$ for a fixed value of $g$, we can get a variational estimate (denoted as $E_{gs}^{var}$) of the ground state energy and an optimal permanent variational wave function $\Phi_e$. In Fig. 2, the variational energy $E_{gs}^{var}$ is compared with the exact ground state energy $E_{gs}^{ext}$ obtained by exact diagonalization (ED) [32]. We see that over the full range of $0 \le g < \infty$, the variational estimates agree with the exact values very well. The discrepancy is apparent only for the exponential-type variational wave function in the region around $g = 4$, with a relative error about 5.6%. However, in this region, the Lorentz-type wave function is a much better approximation, reducing the relative error to about 2%. The general observation is that for $g \lesssim 8$, the Lorentz-type wave function yields a better upper bound for the ground state energy, while for $g \gtrsim 8$, the exponential-type wave function is better, although the latter fact is hardly visible in Fig. 2. Here in passing, we mention that if we take the Gross-Pitaevskii approximation, the orbital occupied by all the particles should be the zero-momentum Bloch state, as this choice minimizes the kinetic energy and the interaction energy simultaneously. The energy per particle would be $-2 + g(L-1)/L$, i.e., linear in $g$, which is qualitatively wrong for large values of $g$.

Other than energy, a more stringent test for the accuracy of the variational wave function is its overlap with the exact ground state. We have thus Fig. 3(a), in which the overlap between the exact ground state $|GS\rangle$, which is obtained by ED, and the energy-minimizing (hence the subscript $e$) VWF $|\Phi_e\rangle$, is shown as a function of $g$. We see that in the full range, the overlap is at least 0.78 for the exponential-type VWF for a system as large as $N = L = 12$, and this number is even higher (0.95) for the Lorentz-type VWF. To appreciate these numbers, one should note that the dimension of the many-body Hilbert space is as large as $1\,352\,078$. We also note that Fig. 3(a) and Fig. 2 are consistent with each other. In Fig. 3(a), all the curves show minima in the proximity of $g = 4$. This is exactly where the variational energies deviate most significantly from the exact one in Fig. 2. In this region, the Lorentz-type VWF has a much higher overlap than the exponential-type VWF with the exact ground state and accordingly, in this region the former has a lower energy as shown in Fig. 2. On the other hand, when $g \gtrsim 8$, the exponential-type VWF be-

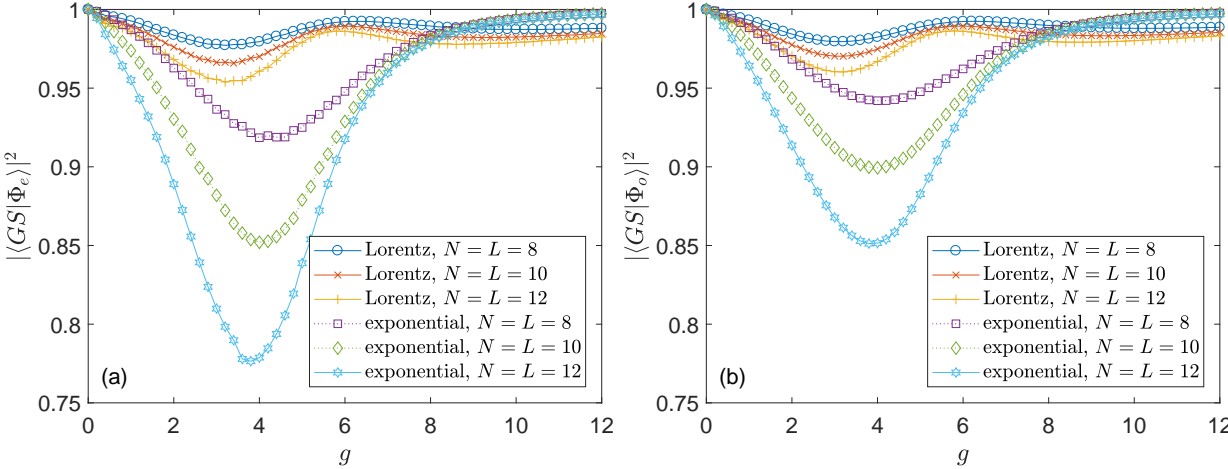

FIG. 3. (Color online) Overlap between (a) the energy-minimizing permanent variational state $\Phi_e$ and (b) the overlap-maximizing permanent variational state $\Phi_o$ constructed with either the Lorentz-type (40a) or the exponential-type (40b) primitive orbitals with the exact ground state $|GS\rangle$. We see that for each lattice size $L$, in the region of $g \lesssim 8$, the Lorent-type variational wave function has a higher overlap with the exact ground state; while in the region of $g \gtrsim 10$, the exponential-type variational wave function becomes better.

comes better by the overlap criterion and accordingly, its energy is lower as shown in Fig. 2. The fact that in the small-$g$ region, the Lorentz-type VWF wins over the exponential-type VWF while in the large-$g$ region, the exponential-type VWF takes over might be understandable in view of the superfluid-Mott insulator transition. The large-$g$ region corresponds to the insulator phase, in which because of the strong particle-particle repulsion, each particle tends to be localized in its own site and the tunneling into neighboring sites should be exponentially small. The small-$g$ region corresponds to the superfluid phase, in which the particles are more mobile and a more extended orbital should be more appropriate.

So far, we have been taking the energy minimizing state $\Phi_e$ among either class of VWFs as an approximation of the exact ground state $|GS\rangle$. Usually, this is the only thing one can do if the exact ground state is unavailable. However, if the exact ground state is available in a certain way, say, by exact diagonalization as we do here, a natural alternate approximation of it should be the variational state having maximal overlap with it. Let us denote it as $\Phi_o$, with the subscript meaning overlap. Hence, we have two related by different optimization problems. One is energy minimization and the other overlap maximization. There is no reason that the two solutions $\Phi_e$ and $\Phi_o$ should be the same, and by definition, we have $|\langle GS|\Phi_e\rangle|^2 \leq |\langle GS|\Phi_o\rangle|^2$ necessarily. In Fig. 3(b), we show $|\langle GS|\Phi_o\rangle|^2$ as a function of $g$. In comparison with Fig. 3(a), we see that all the curves shift upwards as expected. For the exponential-type states, the increase of the overlap is quite apparent. For instance, while in Fig. 3(a), the minimum of $|\langle GS|\Phi_e\rangle|^2$ is about 0.78 for $N = L = 12$, in Fig. 3(b), the minimum of $|\langle GS|\Phi_o\rangle|^2$ is about 0.85. This strongly indicates that the two optimization problems are related but really different. For the Lorentz-type states, the increase of the overlap is

less apparent but still visible. For $N = L = 12$, the minimum of the overlap increases from 0.95 to 0.96. From the curves in Fig. 3, by extrapolation one can infer that even for a system as large as $N = L = 20$, with a many-body Hilbert space of dimension about $6.9 \times 10^{10}$, the minimal values of the overlaps $|\langle GS|\Phi_e\rangle|^2$ and $|\langle GS|\Phi_o\rangle|^2$ would be about 0.9 if we take the Lorentz-type orbital. These numbers are very impressive.

We also note that in Fig. 3, all the curves, regardless of the primitive orbital type or the criterion, show minima in the vicinity of $g = 4$. This should be anticipated in view of the superfluid-Mott insulator transition. According to previous works [33, 34], the transition occurs at about $g_c = 3.61$. Close to the transition, the exact ground state should be most complex and it is hardest to approximate it with some simple functions.

In hindsight, the large overlap between the variational states and the exact ground state should be reasonable. There are at least three reasons that are in favor of such a welcome result. First, both VWFs can reproduce the exact ground state in either limit of $g = 0$ and $g = \infty$. Second, it is well-known that for an arbitrary value of $g$, by the Perron-Frobenius theorem [26], the ground state is strictly positive everywhere in the Fock-state basis. This property is shared by both VWFs by construction. Third, it is also known that the ground state belongs to the trivial representation of the symmetry group of the model (the dihedral group), or more specifically, it is invariant under all translations and reflections, a property again shared by both VWFs by construction.

Finally, as yet another check of the quality of the VWFs, we consider the single-particle correlator $\langle a_0^\dagger a_x \rangle$. This expression is convenient for the exact ground state, which is obtained by ED in the Fock-state basis. For the VWFs, which are in the first-quantization formalism, we

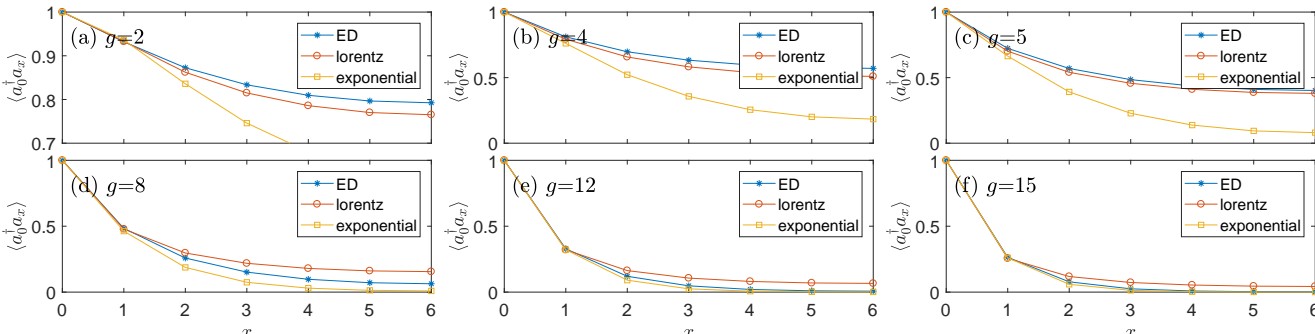

FIG. 4. (Color online) Single-particle correlator $\langle a_0^\dagger a_x \rangle$. In each panel, the $*$ markers are for the exact ground state $|GS\rangle$, while the circles and squares are for the energy-minimizing variational state $\Phi_e$ with Lorent-type or exponential-type orbitals, respectively. The common parameters are $N = L = 12$. Note that because of the periodic boundary condition, the largest possible distance between two sites is 6.

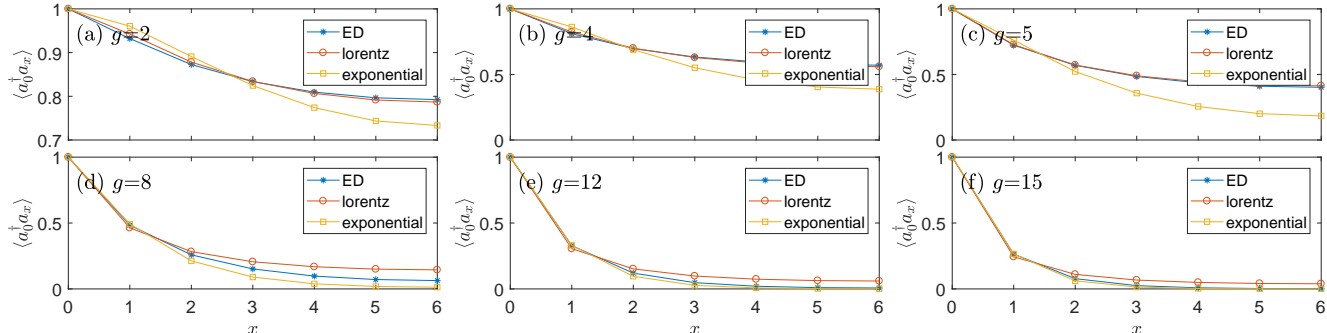

FIG. 5. (Color online) Single-particle correlator $\langle a_0^\dagger a_x \rangle$. In each panel, the $*$ markers are for the exact ground state $|GS\rangle$, while the circles and squares are for the overlap-maximizing variational state $\Phi_o$ with Lorent-type or exponential-type orbitals, respectively. The common parameters are $N = L = 12$. Note that because of the periodic boundary condition, the largest possible distance between two sites is 6.

note that the corresponding single-particle operator $\hat{C}$ has matrix elements $\langle m|\hat{C}|n\rangle = \delta_{m,0}\delta_{n,x}$, and we can use (32) and (33) to calculate its expectation value

$$\langle a_0^\dagger a_x \rangle = \sum_{i,j=1}^{N} \phi_i^*(0)\phi_j(x)\frac{\text{per}(A;i|j)}{\text{per}(A)}. \qquad (42)$$

In Fig. 4, $\langle a_0^\dagger a_x \rangle$ is plotted against $x$ for the energy-minimizing state $\Phi_e$ and the exact ground state $|GS\rangle$. We see a picture consistent with Fig. 2 and Fig. 3. In Fig. 4(a)-(c), when $g$ is small and the lorent-type VWF is better, the correlator predicted by the lorent-type VWF is very close to the exact one. In Fig. 4(e)-(f), when $g$ gets large, the exponential-type VWF is better, and accordingly the correlator predicted by the exponential-type VWF is close to the exact one. For any value of $g$, either the lorent-type or the exponential-type VWF will be a good approximation by all the three criterions. Here it is also interesting to note that while the exponential-type VWF always underestimates the correlator, the lorent-type VWF slightly underestimates it in the small-$g$ region, while overestimates it in the large-$g$ region. This might be related to the superfluid-Mott insulator transition.

In Fig. 5, $\langle a_0^\dagger a_x \rangle$ is plotted against $x$ for the overlap-maximizing states $\Phi_o$. It is normal to expect that $\Phi_o$ reproduces the correlation function better than $\Phi_e$. This is indeed the case. We see that for $g \leq 6$, when $|\langle GS|\Phi_o\rangle|^2$ is significantly higher than $|\langle GS|\Phi_e\rangle|^2$, the curves of $\langle a_0^\dagger a_x \rangle$ shift closer to the exact curve for both types of orbitals. In particular, the curves with the Lorentz-type orbitals almost coincide with the exact curves. For $g \geq 8$, $|\langle GS|\Phi_e\rangle|^2 \simeq |\langle GS|\Phi_o\rangle|^2$, indicating that $\Phi_e \simeq \Phi_o$, we do not see any significant change of the curves.

## IV. OPTIMIZATION ALGORITHM

In the proceeding section, we have seen that for the one-dimensional Bose-Hubbard model with the periodic boundary condition and at unit filling, it is possible to construct some permanent state out of some simple orbitals to approximate its exact ground state very well. This is checked by examining the variational energy, the overlap with the exact ground state, and the single-particle correlation function.

The close approximation is achieved with some *preassigned* orbitals depending on a single parameter $\lambda$. A

natural question is whether the numbers can be further improved by allowing more freedom of the orbitals. This leads to two optimization problems. First, for a given bosonic system with Hamiltonian (28), how can we find a set of orbitals $\{\phi_i, 1 \leq i \leq N\}$, such that the energy expectation value of the permanent state $\Phi$ constructed in (2),

$$E = \frac{\langle \Phi | H | \Phi \rangle}{\langle \Phi | \Phi \rangle}, \tag{43}$$

is minimized? Second, for a given normalized bosonic wave function $|\Psi\rangle$, how can we find the permanent state which is the optimal approximation of it? That is, how can we find the permanent state as in (2) such that the overlap [24, 25]

$$O = \frac{\langle \Phi | \Psi \rangle \langle \Psi | \Phi \rangle}{\langle \Phi | \Phi \rangle} \tag{44}$$

is maximized?

There exists a common simple strategy for both problems [24, 25]. For clarity, let us focus on the first problem for the present. Let us fix $N-1$ orbitals, say, the orbitals $\phi_{2 \leq i \leq N}$, and try to find an optimal $\phi_1$. To this end, we note that with the orbitals $\phi_{2 \leq i \leq N}$ fixed, the numerator and denominator in (43) are both hermitian forms of $\phi_1$. That is, one can find operators $\hat{F}$ and $\hat{G}$ such that $\langle \phi_1 | \hat{F} | \phi_1 \rangle = \langle \Phi | H | \Phi \rangle$ and $\langle \phi_1 | \hat{G} | \phi_1 \rangle = \langle \Phi | \Phi \rangle$. We then can rewrite the ratio as

$$E = \frac{\langle \phi_1 | \hat{F} | \phi_1 \rangle}{\langle \phi_1 | \hat{G} | \phi_1 \rangle}, \tag{45}$$

By definition, $\hat{F}$ and $\hat{G}$ are hermitian single-particle operators depending on the orbitals $\phi_{2 \leq i \leq N}$. Importantly, $\hat{G}$ is even positive definite as long as $\phi_{2 \leq i \leq N}$ are all nonzero, as by definition $\langle \phi_1 | \hat{G} | \phi_1 \rangle = \langle \Phi | \Phi \rangle \geq 0$, with the equality achieved only if $\phi_1 = 0$. Here we recall Proposition 1, which asserts that $\Phi$ is necessarily nonzero if $\phi_{1 \leq i \leq N}$ are nonzero. It is a straightforward but lengthy calculation to derive the explicit expressions of $\hat{F}$ and $\hat{G}$, so we defer it to the Appendix. Suppose we have prepared the operators $\hat{F}$ and $\hat{G}$ (this is the most time-consuming part of the iteration). The optimal $\phi_1$ that will minimize the ratio in (45) is just the solution of the following generalized eigenvalue problem

$$\hat{F}\phi = \varepsilon \hat{G}\phi \tag{46}$$

corresponding to the smallest eigenvalue $\varepsilon_{min}$, and the minimum of the ratio is just $\varepsilon_{min}$.

Once we have updated $\phi_1$, we can turn to $\phi_2$, and then to $\phi_3$, and so on. However, for convenience of programming, we can just make a circular shift of the orbitals $\phi_i \rightarrow \phi_{i-1}$, and continue to update $\phi_1$. In this process, the variational energy decreases monotonically, and as it is lower bounded by the exact ground state energy, it will definitely converge.

We have to mention that (46) is essentially the self-consistency equation derived by Heimsoth before by the method of performing variational differentiation of abstract Hilbert space vectors [14, 15]. However, hopefully here our different point of view has led to a more compact and transparent formalism.

Now it should be clear that the second problem can be treated similarly. The ratio (44) can be written as

$$O = \frac{\langle \phi_1 | \gamma \rangle \langle \gamma | \phi_1 \rangle}{\langle \phi_1 | \hat{G} | \phi_1 \rangle}, \tag{47}$$

where the single-particle orbital $\gamma$ is defined by the summation or partial contraction

$$\gamma(x) = \sqrt{N!} \sum_{x_2,\ldots,x_N=1}^{L} \Psi(x, x_2, \ldots, x_N) \prod_{i=2}^{N} \phi_i^*(x_i). \tag{48}$$

Once $\gamma$ is calculated (again, this is the most time-consuming part), the optimal $\phi_1$ can be obtained by solving a generalized eigenvalue equation similar to (46), with $|\gamma\rangle\langle\gamma|$ replacing $\hat{F}$.

Naively, the summation in (48) has the complexity of $L^{N-1}N$, as $x_{2 \leq i \leq N}$ run independently from 1 to $L$. However, one should note that $\Psi$ is invariant under permutations of $x_{2 \leq i \leq N}$. Making use of this fact and changing the dummy variables from $x_i$ to $y_{i-1}$, (48) can be written as

$$\gamma(x) = \sqrt{N!} \sum_{\mathbf{y}} \Psi(x, \mathbf{y}) \frac{\mathrm{per}(\phi^*(\mathbf{y}))}{\mathbf{n}(\mathbf{y})!}, \tag{49}$$

where the summation is over the ordered $(N-1)$-tuple $\mathbf{y} \equiv (y_1, y_2, \ldots, y_{N-1})$ with $1 \leq y_1 \leq y_2 \ldots \leq y_{N-1} \leq L$, and $\phi^*(\mathbf{y})$ denotes the $(N-1) \times (N-1)$ matrix with its $i$th row being the $y_i$th row of the $L \times (N-1)$ matrix $(\phi_2^*, \phi_3^*, \ldots, \phi_N^*)$. In the denominator, $\mathbf{n}(\mathbf{y})! \equiv \prod_{i=1}^{L} n_i!$, with $n_i$ denoting the times $i$ appears in $\mathbf{y}$. The computational complexity is now on the order of $\binom{L+N-2}{N-1} 2^{N-2} N$.

## A. The multiconfiguration case

So far, we have assumed a single configuration. For better approximation, one can try $M > 1$ sets of orbitals $\{\phi_i^{(\alpha)}, 1 \leq \alpha \leq M, 1 \leq i \leq N\}$, and let the variational wave function be the sum of the permanent wave functions constructed by each set of orbitals. Specifically,

$$\Phi = \sum_{\alpha=1}^{M} \Phi^{(\alpha)} = \sum_{\alpha=1}^{M} \hat{S}(\phi_1^{(\alpha)}, \ldots, \phi_N^{(\alpha)}). \tag{50}$$

Fixing the orbitals $\phi_{2 \leq i \leq N}^{(\alpha)}$ in each set, the variational energy (43) can be written as

$$\begin{aligned} E &= \frac{\sum_{\alpha,\beta=1}^{M} \langle \Phi^{(\alpha)} | H | \Phi^{(\beta)} \rangle}{\sum_{\alpha,\beta=1}^{M} \langle \Phi^{(\alpha)} | \Phi^{(\beta)} \rangle} \\ &= \frac{\sum_{\alpha,\beta=1}^{M} \langle \phi_1^{(\alpha)} | \hat{F}^{(\alpha\beta)} | \phi_1^{(\beta)} \rangle}{\sum_{\alpha,\beta=1}^{M} \langle \phi_1^{(\alpha)} | \hat{G}^{(\alpha\beta)} | \phi_1^{(\beta)} \rangle}, \end{aligned} \tag{51}$$

where $\hat{F}^{(\alpha\beta)}$ and $\hat{G}^{(\alpha\beta)}$ depend on the fixed orbitals $\phi_{2\leq i\leq N}^{(\alpha)}$ and $\phi_{2\leq i\leq N}^{(\beta)}$, and can be calculated with essentially the same formulae as in the Appendix—Just add the superscript $\alpha$ to the orbitals in the bras and $\beta$ to the orbitals in the kets. It is easily seen that

$$(\hat{F}^{(\alpha\beta)})^\dagger = \hat{F}^{(\beta\alpha)}, \quad (\hat{G}^{(\alpha\beta)})^\dagger = \hat{G}^{(\beta\alpha)}. \qquad (52)$$

Apparently, (51) can be cast in the same form as (45), if we identify $\phi_1$ as the concatenated vector

$$\phi_1 \equiv (\phi_1^{(1)}; \phi_1^{(2)}; \ldots; \phi_1^{(M)}), \qquad (53)$$

which is of length $LM$, and define the block operators $\hat{F} \equiv (\hat{F}^{(\alpha\beta)})$ and $\hat{G} \equiv (\hat{G}^{(\alpha\beta)})$, which are of size $LM \times LM$. The same update and iteration procedures can then be carried out.

Similarly, in the multiconfiguration case, the overlap (44) can be written as

$$O = \frac{\sum_{\alpha,\beta=1}^{M} \langle \phi_1^{(\alpha)} | \gamma^{(\alpha)} \rangle \langle \gamma^{(\beta)} | \phi_1^{(\beta)} \rangle}{\sum_{\alpha,\beta=1}^{M} \langle \phi_1^{(\alpha)} | \hat{G}^{(\alpha\beta)} | \phi_1^{(\beta)} \rangle}, \qquad (54)$$

where the single-particle orbital $\gamma^{(\alpha)}$, like $\gamma$ in (48), is defined by the same summation with $\phi_i^{(\alpha)}$ replacing $\phi_i$. Again, the same strategy as above can be applied.

### B. A pitfall with $N = 2$

A pitfall is to be avoided in implementing the multiconfiguration scheme. In the single-configuration case, the hermitian operator $\hat{G}$ is strictly positive definite as long as the orbitals $\phi_{2\leq i\leq N}$ are nonzero. This is because by definition $\langle \phi_1 | \hat{G} | \phi_1 \rangle = \langle \Phi | \Phi \rangle$, and by Proposition 1, $\Phi$ is nonzero if $\phi_{1\leq i\leq N}$ are nonzero. In contrast, in the multi-configuration case (50), we do not necessarily have $\Phi \neq 0$ even if all the orbitals $\{\phi_i^{(\alpha)}, 1 \leq \alpha \leq M, 1 \leq i \leq N\}$ are nonzero—the configurations could cancel each other out. This happens particularly in the two-particle case of $N = 2$. As an illustration let us consider the two-particle, two-configuration case. For any value of $\{\phi_2^{(1)}, \phi_2^{(2)}\}$, if we choose $\{\phi_1^{(1)}, \phi_1^{(2)}\}$ as $\{\phi_2^{(2)}, -\phi_2^{(1)}\}$,

$$\begin{aligned} \Phi &= \hat{\mathcal{S}}(\phi_1^{(1)}, \phi_2^{(1)}) + \hat{\mathcal{S}}(\phi_1^{(2)}, \phi_2^{(2)}) \\ &= \hat{\mathcal{S}}(\phi_2^{(2)}, \phi_2^{(1)}) - \hat{\mathcal{S}}(\phi_2^{(1)}, \phi_2^{(2)}) = 0. \end{aligned} \qquad (55)$$

That the total wave function $\Phi$ could vanish means that $\hat{G}$ in the multi-configuration case is just semi-positive definite. It could have zero eigenvalues. Theoretically, this does not cause any problem because when the denominator of (51) vanishes, its numerator vanishes too. In other words, the eigenvectors of $\hat{G}$ with the zero eigenvalue are also eigenvectors of $\hat{F}$ with the zero eigenvalue. In numerics, one has to restrict $\hat{F}$ and $\hat{G}$ to the subspace spanned by the eigenvectors of $\hat{G}$ with nonzero (hence positive) eigenvalues. This can be easily implemented once $\hat{G}$ is diagonalized.

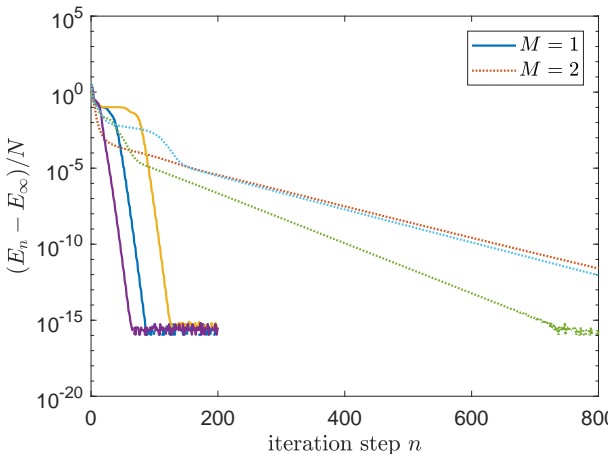

FIG. 6. (Color online) Convergence of the energy of the permanent variational state. Each curve corresponds to a different set of initial orbitals. The solid (dotted) lines are with $M = 1$ ($M = 2$) configuration(s). In each step, one orbital is updated. The energy after $n$ steps is denoted as $E_n$. The limiting value $E_\infty$ is approximated by $E_{2400}$. The setting is a one-dimensional Bose-Hubbard model in a harmonic trap with the Hamiltonian (56). The parameters $(N, L, \kappa, g) = (3, 13, 0.2, 5)$.

In practice, this cautious extra effort is necessary only for $N = 2$. In our extensive numerical simulations, we have never encountered a case of $\hat{G}$ becoming singular for $N \geq 3$. The reason is yet to be understood but it is not surprising as when $N \geq 3$, for fixed $\phi_{2\leq i\leq N}^{(\alpha)}$, unlike (55), it is hard to find $\phi_1 = (\phi_1^{(1)}; \phi_1^{(2)}; \ldots; \phi_1^{(M)})$ to make $\Phi$ vanish; likely there is no solution. But for $N = 2$, it occurs necessarily. With $M$ configurations, generally $\hat{G}$ has $\binom{M}{2}$ eigenvectors with the eigenvalue zero. This number can be understood in view of (55). For fixed orbitals $\phi_{2\leq i\leq N}^{(\alpha)}$, one can find $\binom{M}{2}$ linearly independent vectors $\phi_1$ such that $\Phi$ vanishes.

### C. Convergence of the algorithm

We take a concrete model to illustrate the convergence behavior of the algorithm. Consider a one-dimensional Bose-Hubbard model in a harmonic trap. The Hamiltonian is

$$\begin{aligned} H_{trap} &= -\sum_{x=-L_0}^{L_0-1} (a_x^\dagger a_{x+1} + \text{h.c.}) + \frac{g}{2} \sum_{x=-L_0}^{L_0} a_x^\dagger a_x^\dagger a_x a_x \\ &\quad + \kappa \sum_{x=-L_0}^{L_0} x^2 a_x^\dagger a_x. \end{aligned} \qquad (56)$$

Here $\kappa > 0$ is the stiffness of the harmonic potential. For symmetry, we have assume a lattice of size $L = 2L_0 + 1$. We take the open boundary condition.

We start from $MN$ random orbitals [see Eq. (50)], where $M$ is the configuration number and $N$ is the par-

ticle number. Essentially, we just generate an $ML \times N$ matrix with each element chosen randomly from the interval $[0, 1]$ according to the uniform distribution. The orbitals (the columns) are then updated in a circular way. The variational state after $n$ updates (steps) will be denoted as $\Phi_n$, and its energy will be denoted as $E_n$. Note that $E_n$ is obtained simultaneously in solving the generalized eigenvalue problem (46).

By construction, $E_n$ decreases monotonically and will definitely converge. The concern is in which way and with what rate it converges to its limit $E_\infty$. This is studied in Fig. 6, where for a set of values of the parameters $(N, L, \kappa, g)$ and with $M = 1$ or $2$, some typical trajectories of $E_n - E_\infty$ (here we just approximate $E_\infty$ by some $E_n$ with a large enough $n$) are displayed. For each value of $M$, we have three trajectories corresponding to three different sets of initial orbitals.

We see that often the trajectory is not very regular— It is neither pure exponential nor pure power law, but clearly divides into different parts. In many cases, after some relaxation or transition stage, which can last for a long time, the energy eventually enters an exponentially decreasing mode. In our extensive numerical simulations, the observation is that the trajectory of the energy depends not only on the model and the model parameters, but also on the initial conditions and can differ significantly from run to run. We cannot draw any definite rule for the convergence rate of the energy, but the feeling is that the convergence tends to be slower with more configurations and weaker interactions.

### 1. The non-interacting case

Possibly the most embarrassing and surprising thing is that the convergence is slowest in the non-interacting limit. In this trivial case, the exact ground state $|GS\rangle$ is simply a condensate-type (and hence a permanent) state with all particles occupying the single-particle ground state. We do not need to invoke the algorithm for its calculation, however, if we do, the convergence is as slow as a power law. In Figs. 7(a) and 7(b), with $g = 0$ but the other parameters the same as in Fig. 6, trajectories of the energy error $E_n - E_\infty$ and the infidelity $1 - |\langle GS|\Phi_n\rangle|^2$ are shown respectively for a particular run. Here we take $E_\infty$ to be the exact ground state energy. In either figure, the curve drops down steeply at about $n \sim N$, and afterwards it follows a straight line in the log-log plot. Basically, the picture is that after the first round of update, i.e., when each orbital has been updated once, the variational wave function is already very close to the exact state (the overlap is over 0.99 in the particular case). Afterwards, it improves slowly by a power law.

The exact reason behind the slow convergence is yet to be understood. Here we just emphasize that the convergence is slow only in the asymptotic sense. In the initial phase, the algorithm can already deliver the state $\Phi_n$ close enough to its limit $\Phi_\infty = |GS\rangle$.

In Figs. 7(c) and 7(d), we show two snapshots of the constituent orbitals, the particle density distribution $\rho(x, x)$, and the correlation function $\rho(0, x)$, which are respectively the diagonal and off-diagonal parts of the single-particle reduced density matrix $\rho$ defined as

$$\rho(x_1, x_2) = \langle a_{x_2}^\dagger a_{x_1} \rangle. \tag{57}$$

We see that even for an $n$ as small as $n = N = 3$, the permanent variational state $\Phi_n$ can already reproduce the exact values of $\rho(x, x)$ and $\rho(0, x)$ to high precision. The intriguing thing is that while the total wave function is already very close to the exact many-body ground state in terms of energy, overlap, and some most relevant correlation functions, the constituent orbitals are still far away or at least visibly different from the exact single-particle orbital. In view of Proposition 2, which states that different sets of orbitals necessarily result in different many-body wave functions, the current observation implies that the latter is not necessarily very sensitive to perturbations of the former in some cases. This in turn implies that it might not be a good idea to use convergence of the orbitals as a criterion in determining the termination of the algorithm [15].

### 2. Local minima

It should be no wonder that the algorithm can get stuck in a local minimum like many other greedy algorithms. This is illustrated in Fig. 8(a) with a model of (56) and some specific value of $g$. We see that many trajectories of $E_n$ settle down on a secondary minimum. Note that for clarity, we have shown only the first 100 steps, but actually the horizontal lines extend all the way up to $n = 1500$.

In Fig. 8(b), we plot the possible eventual values of the variational energy $E_n$ and the overlap $|\langle GS|\Phi_n\rangle|^2$ as functions of $g$. For each value of $g$, like in Fig. 8(a), we have run the algorithm 20 times, each time up to $n = 1500$. We see that for $g$ smaller than some critical value $g_c \simeq 8$, we get only a single value for either of $E_\infty$ and $|\langle GS|\Phi_\infty\rangle|^2$, which indicates that there is only a global minimum, however for $g$ larger than $g_c$, we get two different values for either of $E_\infty$ and $|\langle GS|\Phi_\infty\rangle|^2$, which indicates the presence of a second, local minimum.

Our experience is that local minima are ubiquitous. Generally, their number increases with the number $M$ of configurations. To avoid them and enhance the probability of hitting the global minimum, we simply run the algorithm multiple times, say 20 times for $M = 4$ configurations.

### 3. Real versus complex

So far, we have assumed the optimal orbitals to be real. For many systems with the time reversal symmetry, the many-body ground state is real and the assumption

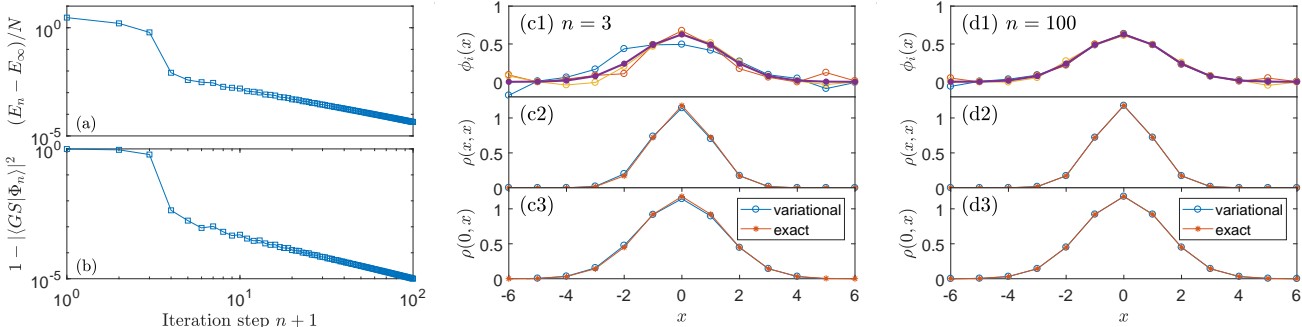

FIG. 7. (Color online) (a) and (b): Trajectories of the energy error $E_n - E_\infty$ and the infidelity $1 - |\langle GS|\Phi_n\rangle|^2$ in a particular run in the non-interacting case ($g = 0$). (c) and (d): The orbitals $\phi_i(x)$, the particle density distribution $\rho(x,x)$, and the one-particle correlator $\rho(0,x)$ at two snapshots. In (c1) and (d1), the solid dots represent the single-particle ground state. The setting is the same as in Fig. 6. The parameters are also the same except for $g$.

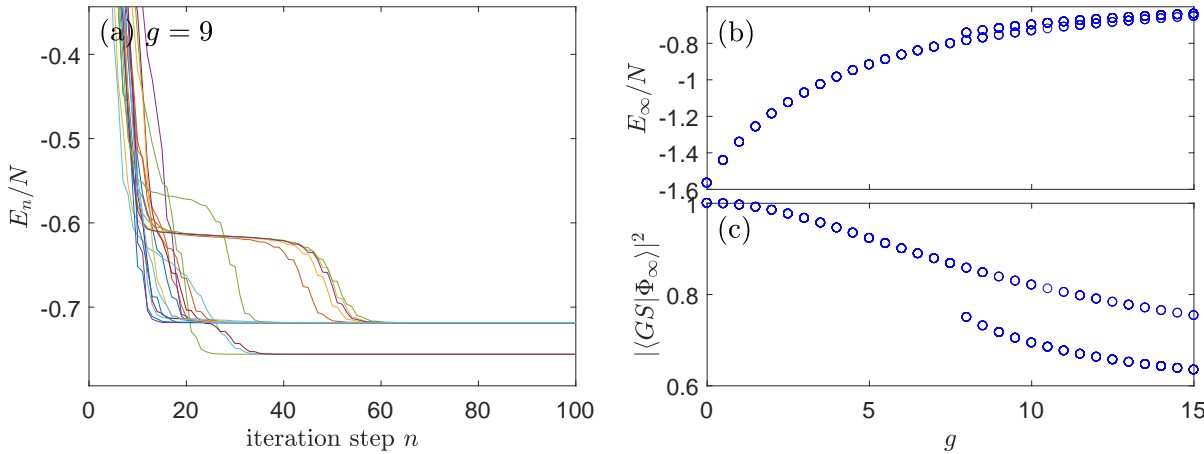

FIG. 8. (Color online) (a) Twenty trajectories of the variational energy $E_n$. Each trajectory starts with a different set of randomly generated initial orbitals. (b) Possible limiting values of the variational energy $E_\infty$ and the overlap $|\langle GS|\Phi_\infty\rangle|^2$. Note that for each value of $g$, we have 20 runs as in (a) and in each run the number of iteration steps is 1500. The setting is a one-dimensional Bose-Hubbard model in a harmonic trap with the Hamiltonian (56). The fixed parameters are $(N, L, \kappa, M) = (3, 13, 0.2, 1)$.

that the optimal orbitals should also be real seems very reasonable. However, this is not the case.

We take a minimal model to illustrate the possibility that the optimal orbitals could be complex although the total wave function is real. Consider a two-particle, two-site Bose-Hubbard model, i.e., a model with $(N, L) = (2, 2)$. The Hamiltonian is

$$H_{ds} = -(a_1^\dagger a_2 + a_2^\dagger a_1) + \frac{g}{2}(a_1^\dagger a_1^\dagger a_1 a_1 + a_2^\dagger a_2^\dagger a_2 a_2). \quad (58)$$

Consider a real wave function $\Psi(x_1, x_2)$, with $x_{1,2} = 1, 2$, of this model. By Proposition 3 or Proposition 4, $\Psi$ can be written as a permanent state

$$\Psi = \hat{S}(u, v) \quad (59)$$

with two orbitals $u$ and $v$. The question is whether $u$, $v$

can both be real. Componentwise, (59) means

$$\Psi(1, 1) = \sqrt{2}u_1 v_1, \quad \Psi(2, 2) = \sqrt{2}u_2 v_2,$$
$$\Psi(1, 2) = \frac{1}{\sqrt{2}}(u_1 v_2 + v_1 u_2).$$

We have then

$$\Delta \equiv \Psi(1, 2)^2 - \Psi(1, 1)\Psi(2, 2) = \frac{1}{2}(u_1 v_2 - v_1 u_2)^2.$$

We thus see that for (59) to have real solutions, a necessary condition is that the quantity $\Delta$ be non-negative. It is easy to verify that this is also sufficient.

Hence, when $\Delta < 0$, in the single configuration approximation, the wave function $\Psi$ can be exactly recovered with complex orbitals but not with real orbitals. Note that such a condition is satisfied by a cat-type state, which can be realized as the ground state of the model (58) with an attractive on-site interaction $g < 0$. In Fig. 9, we show the variational energies of the ground

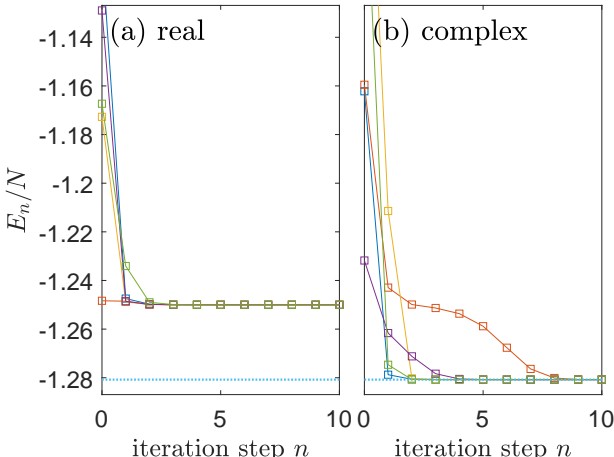

FIG. 9. (Color online) Trajectories of the energy of the permanent variational state with either (a) real or (b) complex orbitals. In each case, we have five runs starting with five different sets of random orbitals. The number of configurations is $M = 1$. The horizontal dotted lines indicate the exact ground state energy. The setting is a two-particle, two-site Bose-Hubbard model with the Hamiltonian (58). The parameter $g = -1$.

state of such a model, calculated with either real or complex orbitals. We see that while complex orbitals can deliver the exact value, real orbitals miss it with some overestimation.

Therefore, we see that at least theoretically, complex orbitals are superior to real orbitals for energy minimization. However, the observation is that for all the models we consider in this paper, if the interaction is repulsive ($g \geq 0$) and if we take the single configuration approximation ($M = 1$), the complex approach leads to identical results with the real approach. More specifically, in these circumstances, even if we start with complex orbitals, the iteration will result in real orbitals. The reason is yet to be understood. For multiple configurations ($M > 1$), with complex orbitals, often we do get lower energies, however, the improvement is not very significant. We thus often confine ourselves to real orbitals in the following. Anyway, real arithmetics are four times fast than complex arithmetics, and the disadvantage in accuracy can be compensated by including more configurations.

## V. APPLICATION OF THE ALGORITHM

### A. The Bose-Hubbard model revisited

With the numerical optimization algorithm above, we can handle more general models. But let us start from the Bose-Hubbard model with the periodic boundary condition and at unit filling, and see how much it can improve over the results in Sec. III.

For this particular model, we take a single configuration ($M = 1$). We never encounter any local minimum,

and the symmetry of the model is perfectly preserved by the optimal orbitals. That is, although we always start from random orbitals, the orbitals we eventually get are always of the same shape and differ from each other just by translations, as described by (38).

In Fig. 10, we show the estimated ground state energy $E_{gs}$ and the overlap $|\langle GS|\Phi_e\rangle|^2$ obtained with unrestricted orbitals. For comparison, also shown are the results with Lorentz or exponential orbitals. We see that in the region $g \leq 6$, the VWF with unrestricted orbitals does not improve much over the VWF with Lorentz orbitals neither by energy nor by overlap. Accordingly, the predicted correlation function $\langle a_0^\dagger a_x \rangle$ is close to that predicted by the Lorentz VWF, as can be seen by comparing Figs. 11(a)-(c) with Figs. 4(a)-(c). However, in the region $g \geq 6$, unrestricted orbitals do lead to improvement over both the Lorentz and the exponential orbitals, both by the criterions of energy and overlap. For instance, at $g = 8$ and with $N = L = 12$, the overestimate in energy (difference between the variational and exact ground state energy) reduces from 0.0127 (Lorentz) and 0.0106 (exponential) to 0.0039 (unrestricted), and simultaneously the deficiency in overlap $(1 - |\langle GS|\Phi_e\rangle|^2)$ reduces from 0.0213 (Lorentz) and 0.0203 (exponential) to 0.0026 (unrestricted). These numbers indicate that the algorithm produces really good approximation of the exact ground state. Indeed, as Figs. 11(d)-(f) show, now the VWF-predicted correlation function $\langle a_0^\dagger a_x \rangle$ almost coincides with the exact values for $g \geq 8$.

Overall, in Figs. 10 and 11, we see that for a system as large as $N = L = 12$ and in the whole range of $g$, the ground state can be very well approximated by a permanent state. This is very impressive in view of the dimension of the many-body Hilbert space, which is as large as $\mathcal{D} = 1\,352\,078$. The exact ground state is obtained by exact diagonalization and is a vector of this size with the Fock states as a basis. In contrast, the permanent variational state is constructed with $N = 12$ orbitals, each of which is a vector of size $L = 12$. That the orbitals are not even independent but related to each other by translations means that the permanent state is encoded with a $12 \times 12$ circulant matrix with only 12 independent variables. From the data compression point of view, with the permanent state as an approximation of the exact ground state, the compression ratio is very high while the fidelity is still very good.

### B. More general models

We now turn to more general models.

First of all, let us break the periodic boundary condition of the Bose-Hubbard model above and replace it with the open boundary condition. The Hamiltonian is

$$H_{obc} = - \sum_{x=-L_0}^{L_0-1} (a_x^\dagger a_{x+1} + \text{h.c.}) + \frac{g}{2} \sum_{x=-L_0}^{L_0} a_x^\dagger a_x^\dagger a_x a_x \quad (60)$$

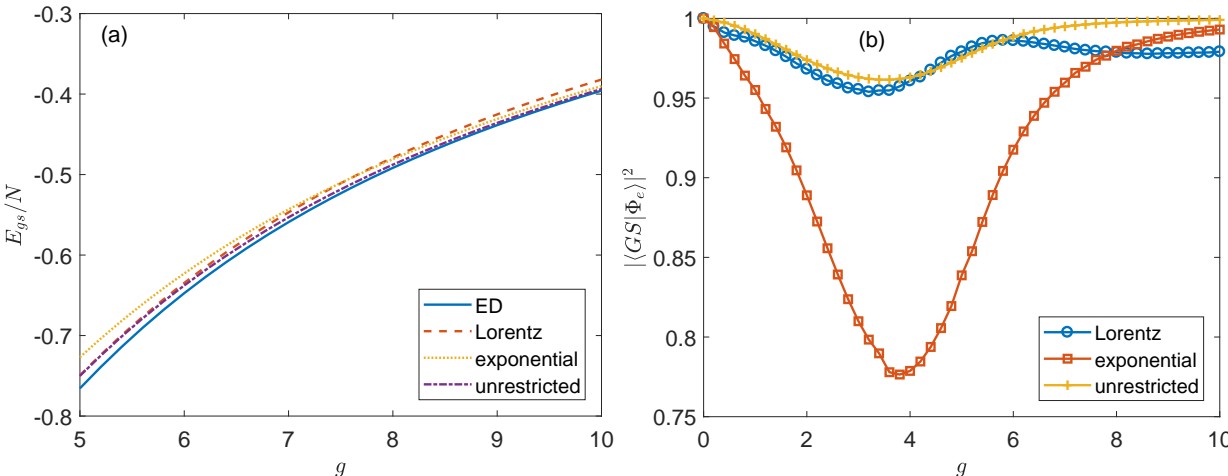

FIG. 10. (Color online) (a) Energy per particle of the energy-minimizing permanent states $\Phi_e$ constructed with either the Lorentz-type (40a) or the exponential-type (40b) primitive orbitals, or unrestricted orbitals. The solid line indicates the exact ground state energy calculated by exact diagonalization (ED). (b) Overlap between the energy-minimizing permanent states $|\Phi_e\rangle$ and the exact ground state $|GS\rangle$. In this figure, the parameters are $N = L = 12$.

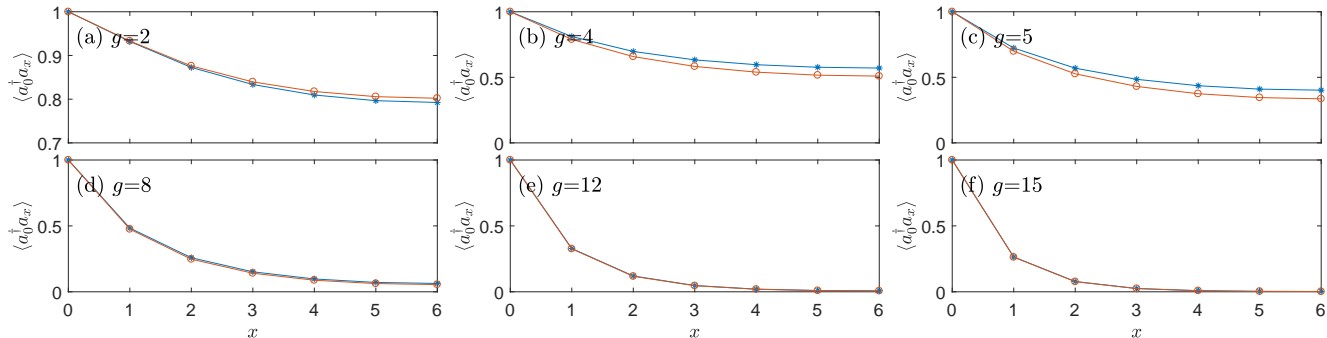

FIG. 11. (Color online) Single-particle correlator $\langle a_0^\dagger a_x \rangle$. In each panel, the $*$ markers are for the exact ground state $|GS\rangle$, while the circles are for the unrestricted energy-minimizing VWF $\Phi_e$. The common parameters are $N = L = 12$. Note that because of the periodic boundary condition, the largest possible distance between two sites is 6.

We still assume unit filling, so the particle number $N = L = 2L_0 + 1$. By the mere change of the boundary condition, the translation symmetry is lost and now the orbitals should differ in shape. It is then unclear what orbitals to choose to construct the permanent state—We have to resort to the numerical algorithm.

In Fig. 12(a), with $N = L = 11$, the permanent estimated ground state energy is compared with the exact diagonalization result, and in Fig. 12(b), the overlap between the permanent variational state $\Phi_e$ and the exact ground state $|GS\rangle$ is shown. We see that like the periodic boundary condition case, across the full range of the on-site interaction $g$, the permanent state is a very good approximation of the exact ground state. The overlap is as large as 0.964 even in the worst case, and the relative error in energy is at most 2%.

In Fig. 13, for three different values of $g$, we show the constituent orbitals, the density distribution function $\rho(x, x)$ and the correlation function $\rho(0, x)$. For each value of $g$, we start from $N$ random orbitals, and then update each orbital 100 times. In the top panels, we see that besides the bulk orbitals which are similar to each other in shape, there are two edge orbitals, which are maximal on the edges and decay into the bulk. As the permanent state is very close to the exact ground state by overlap, these orbitals provide a very good picture of the exact ground state. We also see that the permanent state predicted values of the density distribution and the correlation function agree with the exact results very well.

As a second example, let us consider the one-dimensional Bose-Hubbard model in a harmonic trap, with the Hamiltonian of (56). In Fig. 14, we show the variational ground state energy $E_{gs}$ and the overlap $|\langle GS|\Phi_e\rangle|^2$, calculated with various numbers of configurations and with either real or complex orbitals, as functions of $g$. We see that in the single configuration case ($M = 1$), the real and complex approaches agree with each other exactly. The discrepancy between the variational energy and the exact value is apparent, and the

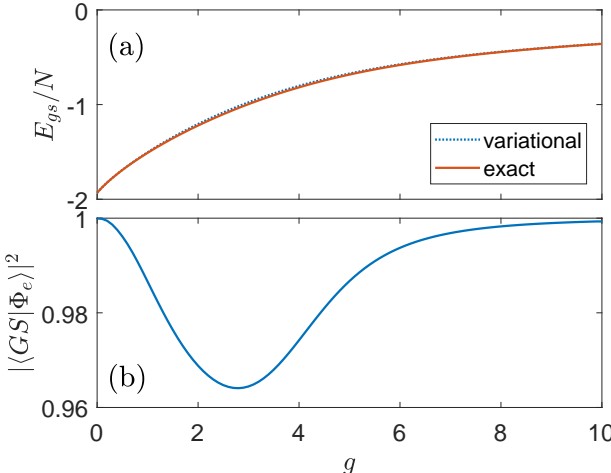

FIG. 12. (Color online) (a) Ground state energy (per particle) of a Bose-Hubbard model at unit filling with the open boundary condition [see (60) for the Hamiltonian]. The solid line is obtained by exact diagonalization, while the dotted line by the permanent variational approach. (b) Overlap between the exact ground state $|GS\rangle$ and the energy-minimizing permanent state $\Phi_e$. The parameters are $N = L = 11$.

overlap drops to 0.76 at $g = 15$. However, the situation improves dramatically if we take $M = 3$ configurations. With three configurations, both the real and the complex approaches get the ground state energy so accurate that the difference with the exact value is hardly visible in the figures. Accordingly, the overlap $|\langle GS|\Phi_e\rangle|^2$ is very close to unity throughout the range of $g$. Actually, the minimal value of the overlap is as large as 0.995 in the two figures. This means that with three configurations, be the orbitals real or complex, we can recover the exact ground state to very high precision. In the intermediate case of $M = 2$ configurations, the energy and the overlap are in-between. A peculiarity is that with real orbitals, the curve of the overlap is discontinuous at some point. This is due to the existence of local minima. At the critical point, two local minima change order in energy, or more precisely, the originally global minimum is surpassed by another minimum which was originally just a local one.

In Figs. 15, 16, and 17, which correspond to $M = 1$, $M = 2$, and $M = 3$, respectively, we show the (real) orbitals $\phi_i^{(\alpha)}$, the density distribution $\rho(x,x)$, and the correlation function $\rho(0,x)$ for three different values of $g$. In Fig. 15, we see that the single configuration approximation, in accord with Fig. 14, is quantitatively not very accurate for large values of $g$. However, the orbitals are consistent with the fermionization picture in the large-$g$ limit [35–37]. In Fig. 16, with two configurations, the similarity between the variational results and exact results improves, but the difference is still apparent for $g = 10$ and 15. A notable feature of the two-configuration approximation is that it breaks the parity symmetry of the model—the density distribution $\rho(x,x)$ and the correlation function $\rho(0,x)$ are asymmetric. This

kind of phenomena is quite common in the conventional Hartree-Fock approximation, and is equally so with us. Below we shall see more examples. In Fig. 17, we have $M = 3$ configurations, and now the variational results almost coincide with the exact results, as is anticipated by the energy and overlap information in Fig. 14.

In the three models above, we see that the permanent approach with a very limited number of configurations can yield very accurate results. The observation is that regardless of the strength of the interaction, this is often the case if the system is not very dilute, or more precisely, if the ratio $N/L_{eff}$ is not too small, where by $L_{eff}$ we mean the effective volume of the system, i.e., the volume accessible to the particles. In the Bose-Hubbard model at unit filling, regardless of the boundary condition, the ratio is 1; in the Bose-Hubbard model in a harmonic trap above, although the lattice is of size $L = 13$, by the shapes of the orbitals or the density distribution, we infer that the effective lattice size $L_{eff} \simeq 7$, so the ratio is about 3/7.

Our experience is that the most challenging situation for the permanent variational approach is a dilute gas in the Tonks-Girardeau limit, in which the system is dilute (i.e., $N \ll L_{eff}$) and the on-site interaction $g$ is strong. We consider such a Bose-Hubbard model in Fig. 18, where we have $N = 4$ particles on a flat, open lattice of size $L = 13$. When $g$ is as small as 1, the single-configuration approximation can get the density distribution $\rho(x,x)$ and the correlator $\rho(0,x)$ accurately. As $g$ increases to 2, the error becomes visible but is still small. However, if $g$ further increases to 4, the discrepancy becomes very apparent.

The natural remedy is to take multiple configurations. In Fig. 19, we show how the variational energy and the overlap improve as the number of configurations increases. The improvement is steady but slow in comparison with Fig. 14. At $g = 4$, which is small by the scale of Fig. 14, even with $M = 5$ configurations, the deviation of the variational energy from the exact value and the deviation of the overlap from unity are still visible. In Fig. 20, we show how the situation in Fig. 18(c) improves by taking more and more configurations. We see that unlike the situation in Fig. 17, with $M = 3$ configurations, the variational curves are still manifestly different from the exact ones. Only with $M = 5$, do the two almost coincide.

The trend of the curves in Fig. 19 suggests that for even larger values of $g$, we would need even more configurations to get a good approximation of the exact ground state. In Fig. 21, we examine the case of $g = 15$. For this value of $g$, the exact ground state is already very close to its fermionization limit at $g = +\infty$. We see that the single configuration approximation fails blatantly and it even breaks the parity symmetry of the model. With $M = 5$ configurations, the situation improves but only with $M = 10$ configurations do the variational predicted density distribution and correlation function agree with the exact values very well.

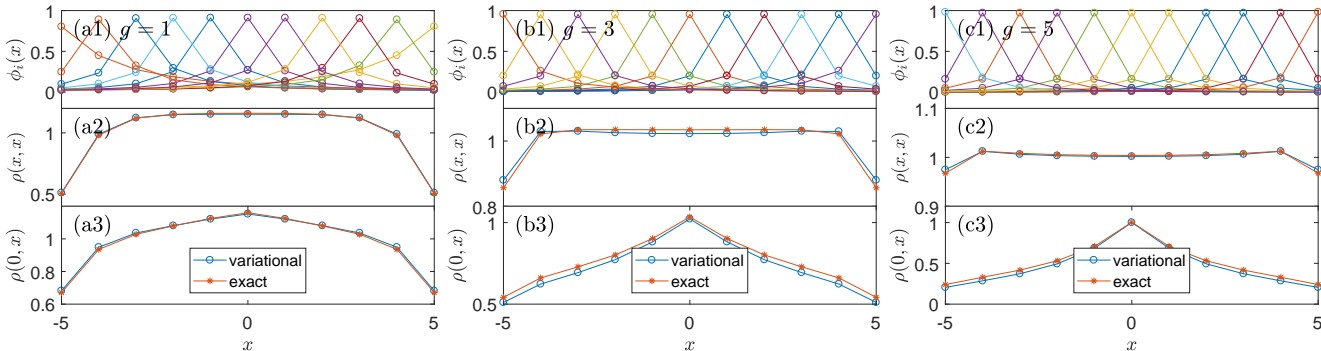

FIG. 13. (Color online) The orbitals $\phi_i(x)$, the particle density distribution $\rho(x, x)$, and the one-particle correlator $\rho(0, x)$ for three different values of $g$. The circles are for the permanent state, while the $*$ markers are for the exact diagonalization results. The setting is a one-dimensional Bose-Hubbard model with the open boundary condition as defined in (60). The fixed parameters are $(N, L) = (11, 11)$.

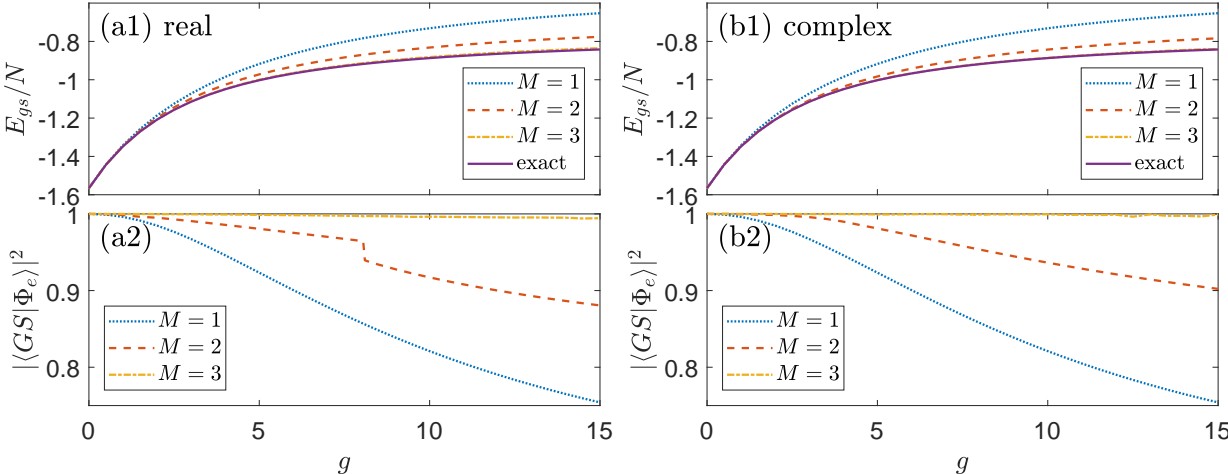

FIG. 14. (Color online) (a1) and (b1): Ground state energy per particle of a Bose-Hubbard model in a harmonic trap. The solid lines are obtained by exact diagonalization, while the other lines are by the permanent variational wave functions with different numbers of configurations. (a2) and (b2): Overlap between the exact ground state $|GS\rangle$ and the energy-minimizing variational state $\Phi_e$. The fixed parameters are $(N, L, \kappa) = (3, 13, 0.2)$. The left (right) column is calculated with real (complex) orbitals.

We thus see that generally taking multiple configurations can effectively reduce the error. The concern is how much price we have to pay. From Sec. IV A, we see that with $M$ configurations, the number of blocks that we have to calculate for preparation of $\hat{F}$ and $\hat{G}$ increases by a factor of $\frac{1}{2}M(M + 1)$. This polynomial growth is mild.

### C. Symmetry breaking and restoration

It is a common observation in the practice of Hartree-Fock approximation that the solution often spontaneously breaks the symmetries of the Hamiltonian [38, 39]. This could also happen with us, as we have seen in Fig. 16 and Fig. 21 above, where the permanent variational states do not respect the $\mathbb{Z}_2$ symmetry of the models.

We have two options to restore symmetry, i.e., to con-struct a state sharing the same symmetry with the exact ground state. The first approach is by brute force, we can simply take more configurations. Hopefully, the ac-curacy of the approximation will improve and the sym-metry is restored alongside. This happens in Fig. 17 and Fig. 21(c). The second approach is based on the group representation theory. We can construct a projection op-erator corresponding to the irreducible representation of the exact ground state, and let it act on the variational state. The resultant state is also a permanent variational state, but generally with more configurations.

Below we take two concrete models to illustrate and compare the two approaches. In the first model, we have $N = 3$ bosons in a double-well potential. The Hamilto-nian is

$$H_{dw} = H_{obc} + \sum_{x=-L_0}^{L_0} V(x) a_x^\dagger a_x, \qquad (61)$$

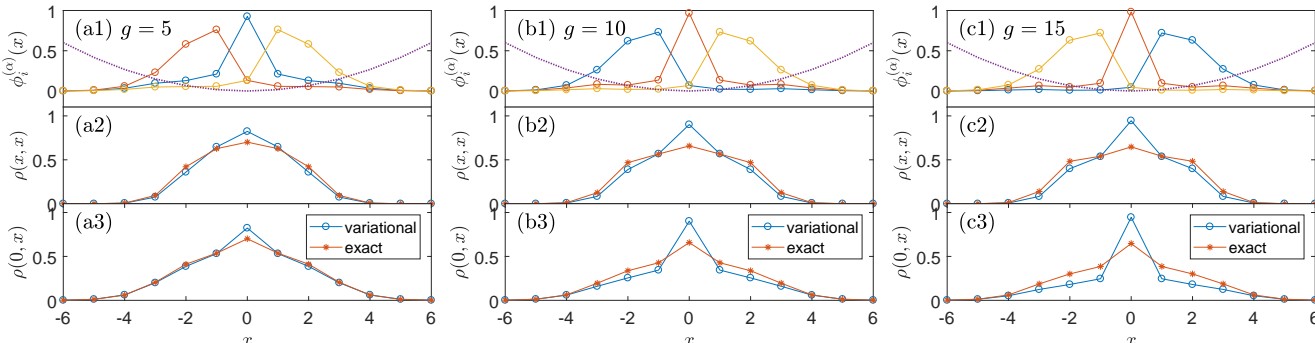

FIG. 15. (Color online) The orbitals $\phi_i(x)$, the particle density distribution $\rho(x,x)$, and the one-particle correlator $\rho(0,x)$ for three different values of $g$. The circles are for the permanent state, while the $*$ markers are for the exact diagonalization results. The setting is a one-dimensional Bose-Hubbard model in a harmonic trap as defined in (56). The fixed parameters are $(N, L, \kappa) = (5, 13, 0.25)$ as in Fig. 6. In the top panels, the dotted line sketches the harmonic potential.

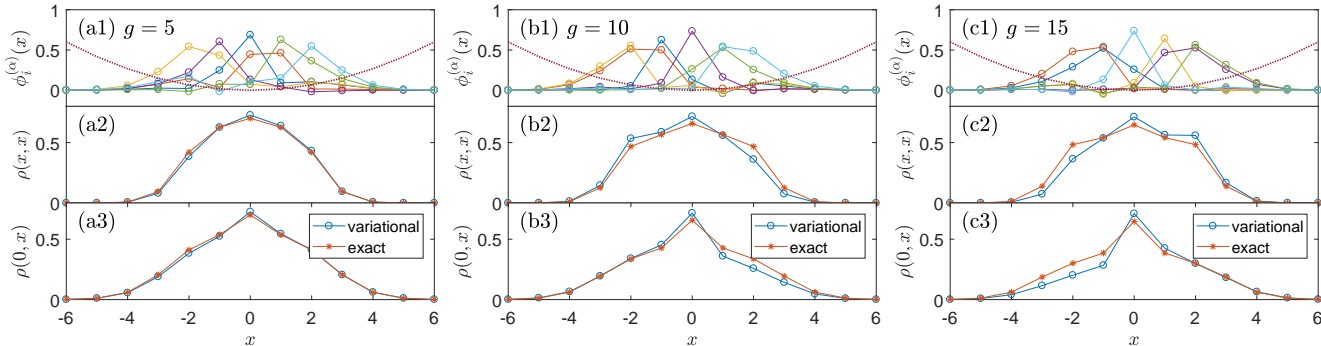

FIG. 16. (Color online) Same as Fig. 15, but with $M = 2$ configurations. Note that the variationally calculated density distribution $\rho(x,x)$ and the correlation function $\rho(0,x)$ are asymmetric.

where the double-well potential $V(x)$ is the superposition of a harmonic trap and a Gaussian bump, i.e.,

$$V(x) = \kappa x^2 + h \exp(-x^2/\sigma^2), \qquad (62)$$

where $\kappa$, $h$, and $\sigma$ are parameters. The system has a $\mathbb{Z}_2$ symmetry, and it can be easily proven that the parity of the ground state is even and consequently the particle density distribution and the correlator are both even functions of $x$. However, in Fig. 22(a1), in the single configuration approximation, we see that the orbitals are apparently asymmetric, with one orbital located in the left well and the rest two in the right well. Consequently, the particle density and the correlator are both asymmetric, and the deviation from the exact values is significant.

To reinstall symmetry by the first approach, we can simply take one more configuration as in Fig. 22(c), where we get symmetric orbitals, and the variational results agree with the exact ones perfectly. We can also try the second approach. Let the permanent state in Fig. 22(a) be $\Phi_e = \hat{\mathcal{S}}(\phi_1, \phi_2, \phi_3)$. An even-parity state can then be easily constructed as

$$\bar{\Phi}_e = \hat{\mathcal{S}}(\phi_1, \phi_2, \phi_3) + \hat{\mathcal{S}}(\hat{P}\phi_1, \hat{P}\phi_2, \hat{P}\phi_3), \qquad (63)$$

where the inversion operator $\hat{P}$ is defined as $(\hat{P}\phi)(x) = \phi(-x)$. The newly constructed variational state $\bar{\Phi}_e$ con-

sists of two configurations, with the new configuration transformed from the old one by inversion. The physical quantities calculated with the projected state $\bar{\Phi}_c$ are shown in Fig. 22(b). We see significant improvement over the pro-projection state $\Phi_e$ in Fig. 22(a). Qualitatively, the symmetry is restored; quantitatively, the overlap with the exact ground state has increased from 0.91786 to 0.99255, and the predicted ground state energy per particle has decreased from 0.98983 to 0.97049 (the exact value is 0.95831). Of course, by construction, we do not expect $\bar{\Phi}_e$ to be optimal in energy among all the two-configuration variational states. Indeed, its energy is slightly higher than that of the optimal state in Fig. 22(c). However, its advantage is that it is obtained for free and is still a fairly good approximation.

The second model is simply the Bose-Hubbard model with the periodic boundary condition. Because of the boundary condition, the lattice can be visualized as a closed lattice ring, and the symmetry group of the model is recognized as that of a regular polygon, i.e., the dihedral group $D_L$. The group consists of $L$ translations (or rotations) and $L$ reflections

$$D_L = \{\hat{S}^m \hat{T}^n, 0 \le m \le 1, 0 \le n \le L - 1\}. \qquad (64)$$

Here the generating operators $\hat{S}$ and $\hat{T}$ are defined as

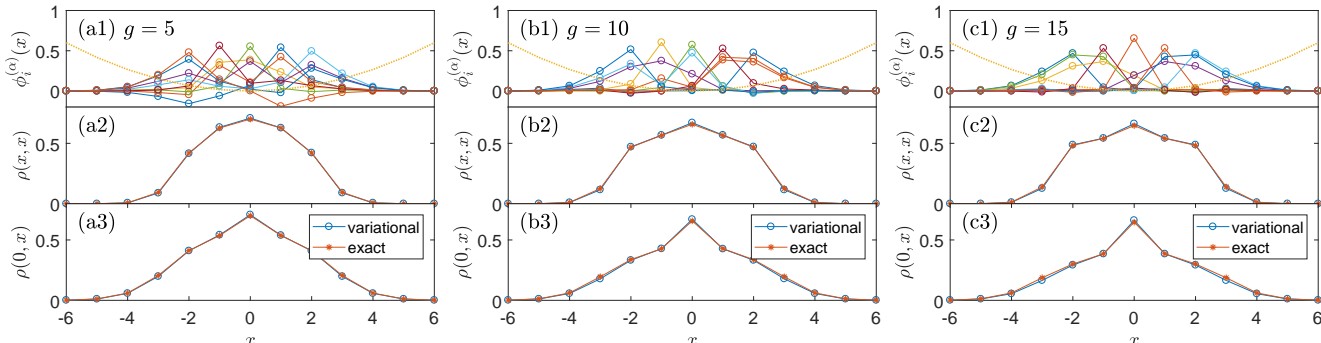

FIG. 17. (Color online) Same as Fig. 15 and Fig. 16, but with $M = 3$ configurations.

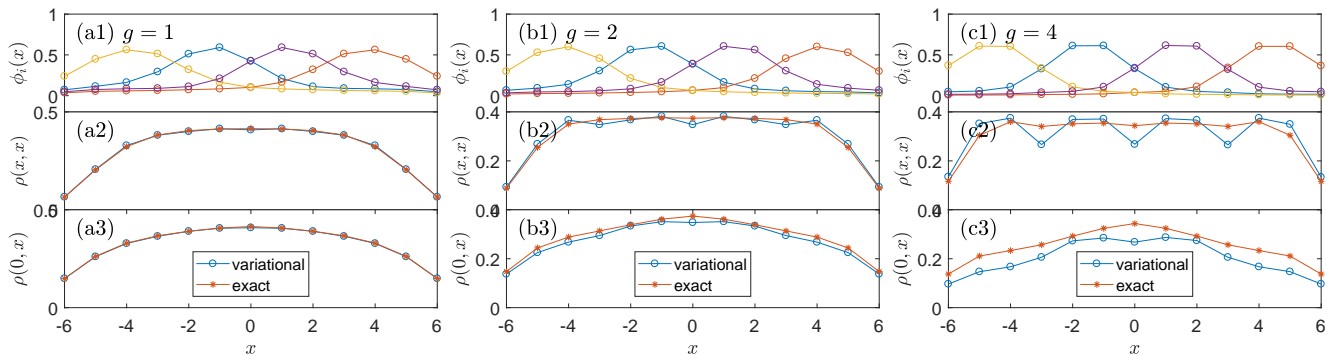

FIG. 18. (Color online) The orbitals $\phi_i(x)$, the particle density distribution $\rho(x,x)$, and the one-particle correlator $\rho(0,x)$ for three different values of $g$. In this figure, we have $N = 4$ bosons on an open chain of $L = 13$ sites.

$(\hat{S}\phi)(x) = \phi(-x)$ and $(\hat{T}\phi)(x) = \phi(x-1)$. By the Perron-Frobenius theorem, the exact ground state $|GS\rangle$ is non-degenerate and positive everywhere in the Fock basis. It then follows easily that $|GS\rangle$ belongs to the trivial representation of the dihedral group. The projection operator for this irreducible representation is simply

$$\hat{\mathcal{P}} = \frac{1}{2L} \sum_{m=0}^{1} \sum_{n=0}^{L-1} \hat{S}^m \hat{T}^n. \tag{65}$$

Therefore, if we obtain an $M$-configuration variational state in the form of (50) by the algorithm, by projection we obtain immediately the following state invariant under all the symmetry transforms of the model,

$$\bar{\Phi}_e = \sum_{m=0}^{1} \sum_{n=0}^{L-1} \sum_{\alpha=1}^{M} \hat{\mathcal{S}}(\hat{S}^m \hat{T}^n \phi_1^{(\alpha)}, \ldots, \hat{S}^m \hat{T}^n \phi_N^{(\alpha)}), \tag{66}$$

which is a $2LM$-configuration state.

In Fig. 23, we consider such a model with $(N, L, g) = (3, 11, 5)$. In the top row, we see that with $M = 1$ to $M = 3$, the variational state always breaks the symmetry of the model completely, i.e., all the translation and reflection symmetries are lost. This is most easily seen from the density distribution. In the bottom row, with the projected states, the symmetry of the exact ground state is recovered. We see that generally, the projection process not only restores the expected symmetry, but also

lowers the energy and increases the overlap. Remarkably, it is most effective in the single configuration case.

## D. When multiple configurations are perfect

In the proceeding sections, we have seen that taking multiple configurations can get us very close to the target state. Here, we discuss the scenario that a generic target state can be exactly recovered by multiple configurations.

By Proposition 4, if the single-particle Hilbert space is of dimension $L = 2$, then for arbitrary $N$, an $N$-boson state is a permanent state, i.e., it consists of a single configuration. For higher values of $L$, a generic state is not a permanent state and a natural question is, at least how many configurations we need to recover it exactly. A quick lower bound is obtained by mere dimension counting. The dimension $\mathcal{D}$ of the many-body Hilbert space is given in (5). By (18), a permanent state has $d = N(L-1) + 1$ degrees of freedom. Hence, we need at least

$$M_0 = \lceil \mathcal{D}/d \rceil \tag{67}$$

configurations to recover a generic state, where $\lceil x \rceil$ denotes the least integer no less than $x$. There is no reason that this lower bound can be achieved. Indeed, it is an underestimate in the special case of $N = 2$. By Proposition 4, we need $\lfloor (L+1)/2 \rfloor$ configurations to recover a

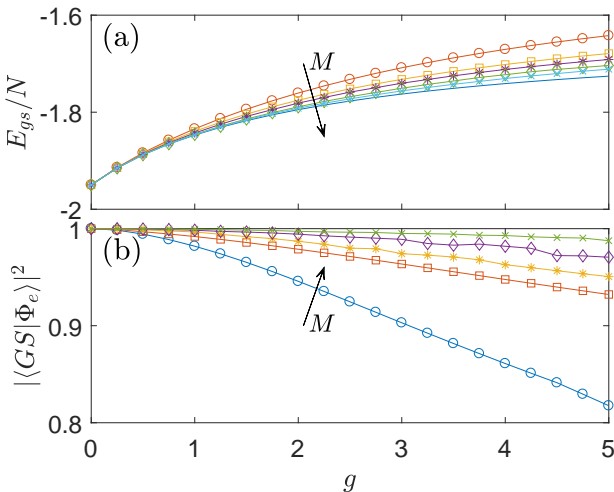

FIG. 19. (Color online) (a) Ground state energy per particle of a Bose-Hubbard model with the open boundary condition. The solid line is obtained by exact diagonalization, while the other lines are by the permanent variational approach with $M = 1$ to $M = 5$ configurations ($M$ increases in the direction of the arrow). The variational calculation is done with real orbitals. (b) Overlap between the exact ground state and the energy-minimizing permanent state. As in Fig. 18, the particle number $N = 4$ and the lattice size $L = 13$.

generic 2-boson state. Here the number scales as $L/2$ for large $L$. However, the estimate of (67) is $M_0 \simeq L/4$ for large $L$.

Although (67) fails for $N = 2$, there are evidences that for many pairs of $(L, N)$, it does give the right answer. Specifically, extensive numerical experiments indicate that that if the pair $(L, N)$ take values among the set of $\{(3, 2), (3, 3), (3, 4), (3, 5), (4, 3)\}$, the $N$-boson wave function can always be written as the summation of $M_0 = 2$ permanent states. Similarly, if $(L, N)$ take values among the set of $\{(4, 4), (5, 3)\}$, the $N$-boson wave function can always be written as the sum of $M_0 = 3$ permanent states.

The numerical experiment is done in the following way. We first generate a random $N$-boson state, then generate a set of $M_0 N$ random single-particle orbitals, with $M_0$ given by (67), and then use the overlap maximization algorithm to update the orbitals. The concern is whether the overlap will surpass the threshold $1 - 10^{-5}$ after 300 rounds of update. If not, a new set of random orbitals are generated and the optimization process is restarted. This process is repeated until in some run the threshold is surpassed. If so, we turn to a new random $N$-boson state and repeat the check.

For the set of values of $(L, N)$ mentioned above, we have checked over $10^5$ random $N$-boson states, and they all passed the check. This is strong evidence that for such $(L, N)$, the naive lower bound of (67) is achieved.

So far, we have failed to find a rigorous proof of the findings above. Here we just reformulate the problem in pure mathematics so that it might be more convenient for

further study. In second quantization, a generic $N$-boson state is

$$\Psi = \sum_{\mathbf{n}} C_{\mathbf{n}} \prod_{j=1}^{L} \left(a_j^\dagger\right)^{n_j} |vac\rangle. \tag{68}$$

Here the summation is over all occupation tuple $\mathbf{n} = (n_1, n_2, \ldots, n_L)$ with $n_j \geq 0$ and $\sum_{j=1}^{L} n_j = N$. By (8), that it can be written as the sum of $M$ permanent states means

$$\Psi = \sum_{\alpha=1}^{M} \prod_{i=1}^{N} \left(\sum_{j=1}^{L} C_{ij}^{(\alpha)} a_j^\dagger\right) |vac\rangle, \tag{69}$$

where $C_{ij}^{(\alpha)}$ are constants. Because the $a_j^\dagger$ operators commute, and the (unnormalized) Fock states $\prod_{j=1}^{L} \left(a_j^\dagger\right)^{n_j} |vac\rangle$ are linearly independent, this is equivalent to saying that the degree-$N$ homogeneous polynomial in $L$ variables

$$P(z_1, z_2, \ldots, z_L) = \sum_{\mathbf{n}} C_{\mathbf{n}} \prod_{j=1}^{L} z_j^{n_j} \tag{70}$$

is expressible as

$$P(z_1, z_2, \ldots, z_L) = \sum_{\alpha=1}^{M} \prod_{i=1}^{N} \left(\sum_{j=1}^{L} C_{ij}^{(\alpha)} z_j\right). \tag{71}$$

Formulated in this way, we see the problem is very similar to the polynomial Waring problem [40]. The difference is just that while in the Waring problem, one seeks to decompose a general degree-$N$ homogeneous polynomial in $N$-th powers of linear forms (linear polynomials), here we seek a decomposition in terms of $N$-th products of linear forms.

## VI. CONCLUSIONS AND OPEN PROBLEMS

We have explored the potential of the permanent state as variational wave functions for bosons. The result is very encouraging. First, we found that for the one-dimensional Bose-Hubbard model with periodic boundary condition and at unit filling, the exact ground state can be well approximated by a permanent state with translation-related orbitals. The permanent state overlaps well with the exact ground state, yields energy close to the exact value, and produces correlation functions close to the exact ones. Then with an iteration algorithm, we examined more general models. It is quite often that the a single permanent state approximates the exact ground state very well, by all the criterions of energy, overlap, and correlation functions. In case the discrepancy is apparent, it can be remedied by including more configurations.

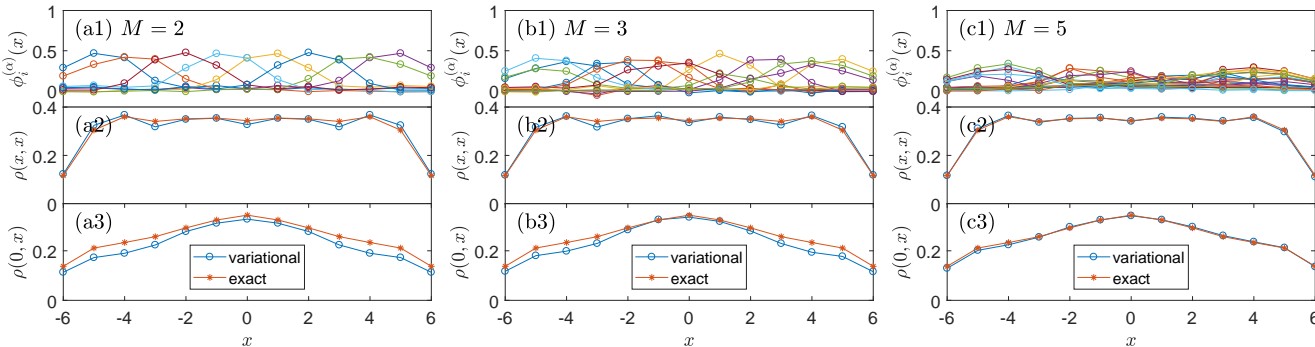

FIG. 20. (Color online) The orbitals $\phi_i^{(\alpha)}(x)$, the particle density distribution $\rho(x,x)$, and the one-particle correlator $\rho(0,x)$ for three increasing values of $M$. The setting and parameters are the same as in Fig. 18(c).

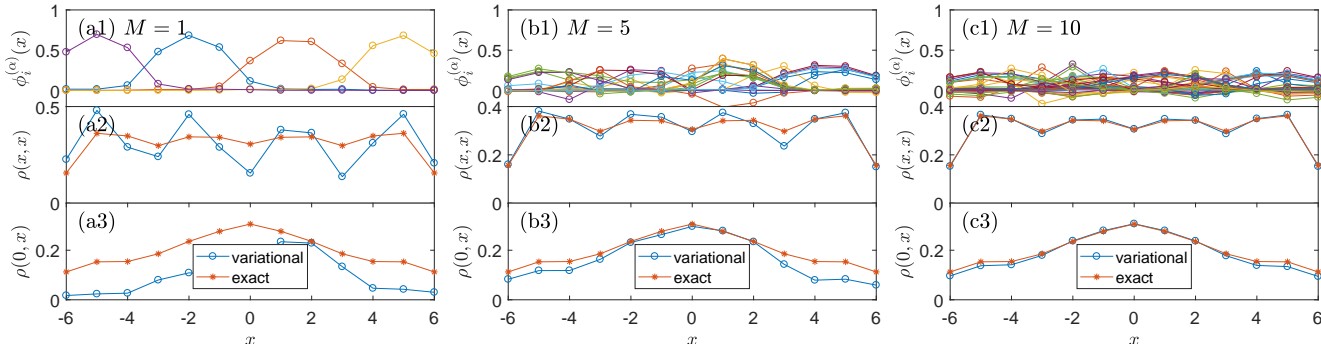

FIG. 21. (Color online) Same as Fig. 20, but the value of $g$ is 15 instead of 4.

The algorithm has its advantages and disadvantages. Let us first address its disadvantages. The primary drawback of a permanent-based approach is of course the permanent computation, which scales unfavorably with the particle number $N$. However, with current computational facilities, it is not prohibitively expensive. On our laptop, in the single-configuration case, it takes about 5.1 seconds to update one orbital for $N = L = 12$, and the time reduces to 1.0 seconds if $N = 10$. Hence, it is totally feasible to study a large enough few-boson system with the algorithm. Note that there is still room for acceleration—computing the permanents of the minors of the overlap matrix can be easily parallelized. While the scaling of the complexity of the algorithm with respect to the particle number $N$ is not that favorable, the scaling with respect to the system volume $L$ is quite favorable. The observation is that for $L \lesssim N$, most time is spent on the permanent calculation and the $L$-dependence is negligible. Only for $L \gg N$, the time needed to update one orbital increases apparently with $L$, but still it increases at most in a polynomial way. For example, for $N = 12$, the time is 5.5, 7.6, and 21.4 seconds for $L = 25, 50, 100$, respectively. Roughly speaking, while we are indeed confined to a limited number of particles, we have essentially no limitation on the system volume, nor the dimensionality of the system. The memory needed by the algorithm is also minimal. While in this paper we have only considered the $(N, L)$ pairs for which the

dimension of the many-body Hilbert space is at most on the order of one million, so that exact diagonalization is possible and we have exact results for comparison, the algorithm can handle other values of $(N, L)$ easily.

In this tentative work, we have only dealt with some one-dimensional lattice models in the Bose-Hubbard category. Generalization to continuum models [41, 42], higher dimensions, and multi-component systems should be straightforward. A potential application or test of the capability of the current variational approach is to search for few-body bound states [43, 44].

Below are some open problems.

The fact that for the one-dimensional Bose-Hubbard model with periodic boundary condition and at unit filling, the exact ground state can be well approximated by a permanent state with translation-related orbitals is very impressive. This should arouse one's interest in the permanent state in its own right. In the field of cold atoms, the folklore is that Bose-Einstein condensation occurs as the temperature lowers, the de Broglie wave lengths of the particles increase, and the wave-packets overlap. Now, a permanent state with translation-related orbitals is in accord with this picture. The overlap between adjacent orbitals can be well adjusted by changing the length scale of the primitive orbital. The concern is, is it possible to realize a transition by tuning the length scale? How does the correlation function depend on the primitive orbital? What is the effect of the dimension-

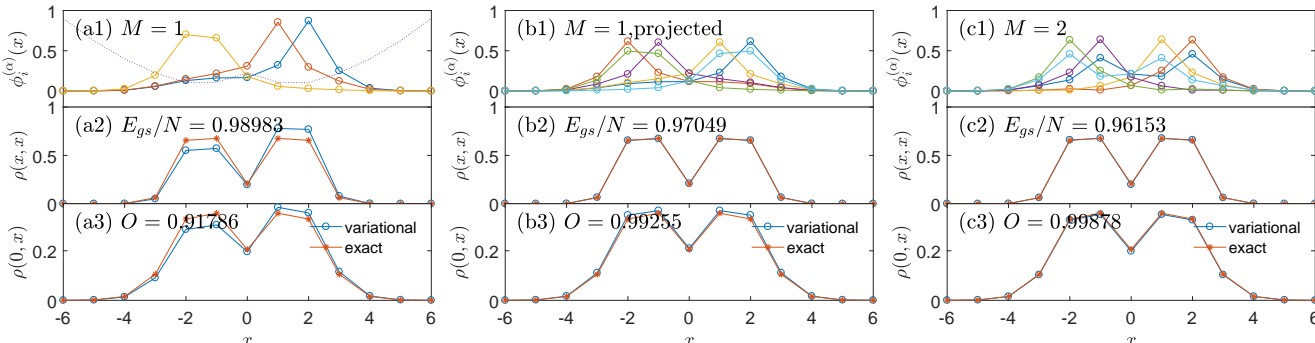

FIG. 22. (Color online) The orbitals $\phi_i(x)$, the particle density distribution $\rho(x,x)$, and the one-particle correlator $\rho(0,x)$ for $N = 3$ bosons in a double-well potential [see Eqs. (61) and (62)]. The parameters are $L = 13$, $g = 2$, $\kappa = 0.5$, $h = 4.5$, $\sigma = 1$. In (a) and (c), the variational state consists of $M = 1$ or $M = 2$ configuration(s), respectively. In (b), the state is built out of the state in (a) by the projection process in (63). In each column, the energy $E_{gs}$ of the variational state and its overlap $O$ with the exact ground state are shown. The exact value of the ground state energy per particle is 0.9583.

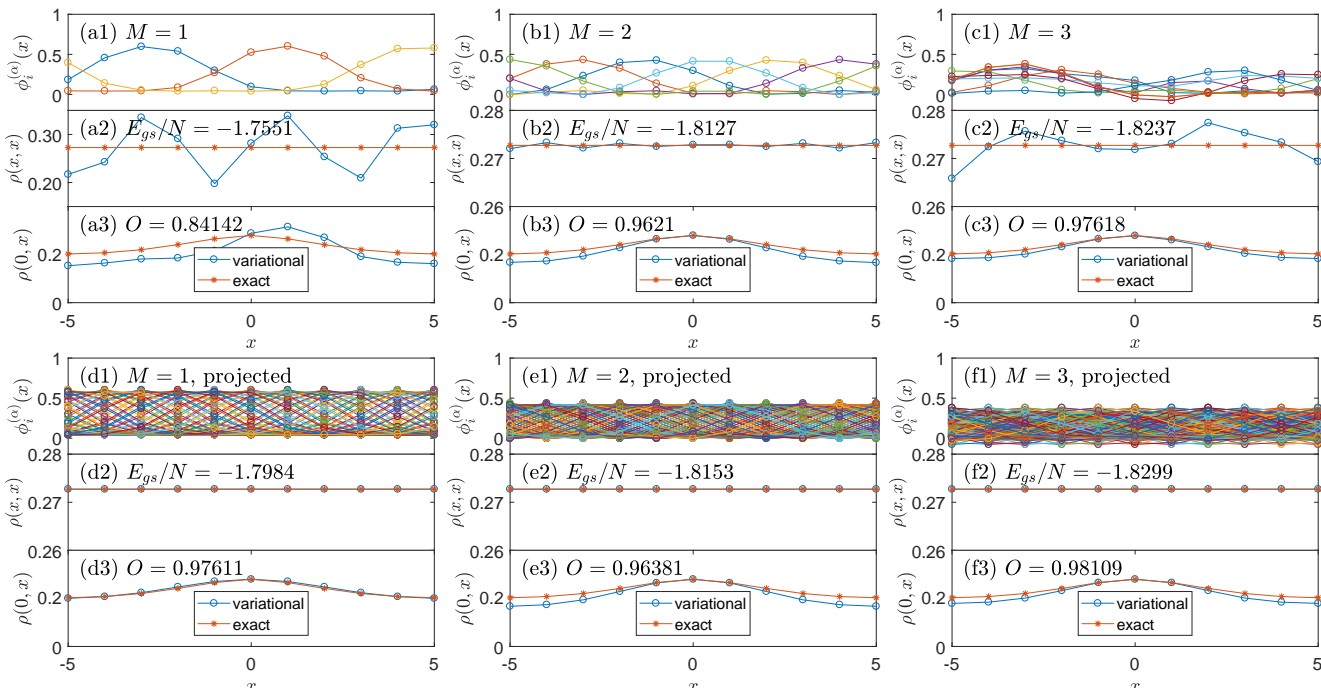

FIG. 23. (Color online) The orbitals $\phi_i(x)$, the particle density distribution $\rho(x,x)$, and the one-particle correlator $\rho(0,x)$ for $N = 3$ bosons on a closed chain with $L = 11$ sites. The on-site interaction strength $g = 5$. The top row corresponds to the pro-projection variational states with $M = 1$ to $M = 3$ configurations. The bottom row corresponds to the projected variational states. As in Fig. 22, for each state, its energy and its overlap with the exact ground state are shown. The exact value of the ground state energy per particle is $-1.8526$.

ality of the lattice? In short, can we use a permanent state with regularly distributed identical wave-packets as a prototypical wave function to model the condensation transition? Note that the problem does not refer to a Hamiltonian.

That a single permanent state is often a very good approximation of the exact ground state motivates two problems. First, is it possible to construct an interacting Hamiltonian [45] whose ground state is exactly a permanent state? Second, how dense are the permanent states in the many-boson Hilbert space? Quantitatively, is there a number $\delta > 0$, such that for any many-boson state there exists a permanent state whose overlap with it is at least $\delta$? Preliminary study suggests that $\delta \geq 0.15$ for $(N, L) = (3, 13)$. This is not a small number in view of the fact that the dimension of the few-body Hilbert space is 455. The ground state of a realistic model is non-generic, so the largest possible value of the overlap of a permanent state with it should be much higher.

In this paper, we have focused on the energy-

minimization problem. The other problem of overlap-maximization, i.e., the problem of finding the optimal single- or multi-configurational permanent approximation of a given bosonic wave function should also be a worthy one. It is about the structure of a bosonic wave function. At least for fermions, it is now well-known that the anti-symmetry condition entails deep structures of the fermionic wave function, with consequences far beyond the commonplace of the Pauli exclusion principle [46–50], and the notion of optimal Slater approximation has proven to be useful in this study [24, 25]. It is fair to expect that for bosons, the symmetry condition also

has far-reaching consequences and hopefully, the notion of optimal permanent approximation is a useful one too.

### ACKNOWLEDGMENTS

The authors are grateful to J. Guo, K. Yang, Y. Xiang and K. Jin for their helpful comments. This work is supported by the Science Challenge Project (NO. TZ2018002) and the Foundation of LCP.

### Appendix: Expressions of $\hat{F}$ and $\hat{G}$ in Sec. IV

Let us start with $\hat{G}$ which is simpler than $\hat{F}$. By an analogy of the Laplace expansion for the determinant, we have

$$
\langle\Phi|\Phi\rangle = \mathrm{per}(A) = \langle\phi_1|\phi_1\rangle\,\mathrm{per}(A;1|1) + \sum_{j_1=2}^{N}\langle\phi_1|\phi_{j_1}\rangle\,\mathrm{per}(A;1|j_1)
$$

$$
= \langle\phi_1|\phi_1\rangle\,\mathrm{per}(A;1|1) + \sum_{j_1=2}^{N}\sum_{i_1=2}^{N}\langle\phi_1|\phi_{j_1}\rangle\langle\phi_{i_1}|\phi_1\rangle\,\mathrm{per}(A;1,i_1|1,j_1). \tag{A.1}
$$

From this expression, we can read off the operator $\hat{G}$ defined by $\langle\phi_1|\hat{G}|\phi_1\rangle = \langle\Phi|\Phi\rangle$. It is

$$
\hat{G} = \mathrm{per}(A;1|1)\hat{I} + \sum_{j_1=2}^{N}\sum_{i_1=2}^{N}\mathrm{per}(A;1,i_1|1,j_1)|\phi_{j_1}\rangle\langle\phi_{i_1}|, \tag{A.2}
$$

where $\hat{I}$ is the identity operator. Note that $\hat{G}$ depends on the orbitals $\phi_{2\leq j\leq N}$ but not on $\phi_1$.

We then turn to $\hat{F}$. For the single-particle part, we have

$$
\langle\Phi|H_1|\Phi\rangle = \sum_{i_1=1}^{N}\sum_{j_1=1}^{N}\langle\phi_{i_1}|K|\phi_{j_1}\rangle\,\mathrm{per}(A;i_1|j_1)
$$

$$
= \langle\phi_1|K|\phi_1\rangle\,\mathrm{per}(A;1|1) + \sum_{j_1=2}^{N}\langle\phi_1|K|\phi_{j_1}\rangle\,\mathrm{per}(A;1|j_1) + \sum_{i_1=2}^{N}\langle\phi_{i_1}|K|\phi_1\rangle\,\mathrm{per}(A;i_1|1)
$$

$$
+ \sum_{i_1=2}^{N}\sum_{j_1=2}^{N}\langle\phi_{i_1}|K|\phi_{j_1}\rangle\,\mathrm{per}(A;i_1|j_1)
$$

$$
= \langle\phi_1|K|\phi_1\rangle\,\mathrm{per}(A;1|1) + \sum_{j_1=2}^{N}\sum_{i_1=2}^{N}\langle\phi_1|K|\phi_{j_1}\rangle\langle\phi_{i_1}|\phi_1\rangle\,\mathrm{per}(A;1,i_1|1,j_1)
$$

$$
+ \sum_{j_1=2}^{N}\sum_{i_1=2}^{N}\langle\phi_1|\phi_{j_1}\rangle\langle\phi_{i_1}|K|\phi_1\rangle\,\mathrm{per}(A;1,i_1|1,j_1) + \sum_{i_1=2}^{N}\sum_{j_1=2}^{N}\langle\phi_{i_1}|K|\phi_{j_1}\rangle\langle\phi_1|\phi_1\rangle\,\mathrm{per}(A;1,i_1|1,j_1)
$$

$$
+ \sum_{i_2\neq i_1,2}^{N}\sum_{j_1\neq j_2,2}^{N}\langle\phi_{i_1}|K|\phi_{j_1}\rangle\langle\phi_1|\phi_{j_2}\rangle\langle\phi_{i_2}|\phi_1\rangle\,\mathrm{per}(A;1,i_1,i_2|1,j_1,j_2). \tag{A.3}
$$

Here in the last line, the summation $\sum_{i_2\neq i_1,2}^{N}$ means that $i_1$ and $i_2$ both run from 2 to $N$, but they must take different values. Similar summation expressions below should be interpreted similarly. We see that the contribution of $H_1$ to

$\hat{F}$ is

$$\text{per}(A;1|1)K + \sum_{j_1=2}^{N}\sum_{i_1=2}^{N} K|\phi_{j_1}\rangle\langle\phi_{i_1}| * \text{per}(A;1,i_1|1,j_1) + \sum_{j_1=2}^{N}\sum_{i_1=2}^{N} |\phi_{j_1}\rangle\langle\phi_{i_1}|K * \text{per}(A;1,i_1|1,j_1)$$

$$+ \sum_{i_1=2}^{N}\sum_{j_1=2}^{N} \langle\phi_{i_1}|K|\phi_{j_1}\rangle\hat{I} * \text{per}(A;1,i_1|1,j_1) + \sum_{i_2\neq i_1,2}^{N}\sum_{j_1\neq j_2,2}^{N} \langle\phi_{i_1}|K|\phi_{j_1}\rangle|\phi_{j_2}\rangle\langle\phi_{i_2}| * \text{per}(A;1,i_1,i_2|1,j_1,j_2). \tag{A.4}$$

Next we turn to the two-particle or the interaction term $H_2$. We have by (34)

$$\langle\Phi|H_2|\Phi\rangle = \frac{1}{2}\sum_{i_1\neq i_2,1}^{N}\sum_{j_1\neq j_2,1}^{N} \langle\phi_{i_1}\phi_{i_2}|U|\phi_{j_1}\phi_{j_2}\rangle \, \text{per}(A;i_1,i_2|j_1,j_2). \tag{A.5}$$

For clarity, three cases will be considered separately. In the first case, two $\phi_1$'s are associated with $U$. We have

$$\frac{1}{2}\sum_{i_2=2}^{N}\sum_{j_2=2}^{N} \langle\phi_1\phi_{i_2}|U|\phi_1\phi_{j_2}\rangle \, \text{per}(A;1,i_2|1,j_2) + \frac{1}{2}\sum_{i_1=2}^{N}\sum_{j_1=2}^{N} \langle\phi_{i_1}\phi_1|U|\phi_{j_1}\phi_1\rangle \, \text{per}(A;1,i_1|1,j_1)$$

$$+ \frac{1}{2}\sum_{i_1=2}^{N}\sum_{j_2=2}^{N} \langle\phi_{i_1}\phi_1|U|\phi_1\phi_{j_2}\rangle \, \text{per}(A;1,i_1|1,j_2) + \frac{1}{2}\sum_{i_2=2}^{N}\sum_{j_1=2}^{N} \langle\phi_1\phi_{i_2}|U|\phi_{j_1}\phi_1\rangle \, \text{per}(A;1,i_2|1,j_1)$$

$$= \sum_{i_2=2}^{N}\sum_{j_2=2}^{N} \langle\phi_1\phi_{i_2}|U|\phi_1\phi_{j_2}\rangle \, \text{per}(A;1,i_2|1,j_2) + \sum_{i_1=2}^{N}\sum_{j_2=2}^{N} \langle\phi_{i_1}\phi_1|U|\phi_1\phi_{j_2}\rangle \, \text{per}(A;1,i_1|1,j_2)$$

$$= \sum_{i_2=2}^{N}\sum_{j_2=2}^{N} \langle\phi_1\phi_{i_2}|U|\phi_1\phi_{j_2}\rangle \, \text{per}(A;1,i_2|1,j_2) + \sum_{i_2=2}^{N}\sum_{j_2=2}^{N} \langle\phi_{i_2}\phi_1|U|\phi_1\phi_{j_2}\rangle \, \text{per}(A;1,i_2|1,j_2). \tag{A.6}$$

Contributions of these expressions to the operator $\hat{F}$ can be easily read off. For example, the operator $\mathcal{O}$ corresponding to the matrix element $\langle\phi_1\phi_{i_2}|U|\phi_1\phi_{j_2}\rangle$ is defined by $\langle\phi_1|\mathcal{O}|\phi_1\rangle = \langle\phi_1\phi_{i_2}|U|\phi_1\phi_{j_2}\rangle$. For the Bose-Hubbard model in which $U$ is an on-site interaction (37), the operator $\mathcal{O}$ is actually a (generally complex) potential with the explicit expression $\mathcal{O}(x) = g\phi_{i_2}^*(x)\phi_{j_2}(x)$.

In the second case, one $\phi_1$ is associated with $U$. We have

$$\frac{1}{2}\sum_{i_2=2}^{N}\sum_{j_1\neq j_2,2}^{N} \langle\phi_1\phi_{i_2}|U|\phi_{j_1}\phi_{j_2}\rangle \, \text{per}(A;1,i_2|j_1,j_2) + \frac{1}{2}\sum_{i_1=2}^{N}\sum_{j_1\neq j_2,2}^{N} \langle\phi_{i_1}\phi_1|U|\phi_{j_1}\phi_{j_2}\rangle \, \text{per}(A;1,i_1|j_1,j_2)$$

$$+ \frac{1}{2}\sum_{i_1\neq i_2,2}^{N}\sum_{j_2=2}^{N} \langle\phi_{i_1}\phi_{i_2}|U|\phi_1\phi_{j_2}\rangle \, \text{per}(A;i_1,i_2|1,j_2) + \frac{1}{2}\sum_{i_1\neq i_2,2}^{N}\sum_{j_1=2}^{N} \langle\phi_{i_1}\phi_{i_2}|U|\phi_{j_1}\phi_1\rangle \, \text{per}(A;i_1,i_2|1,j_1)$$

$$= \sum_{i_2=2}^{N}\sum_{j_1\neq j_2,2}^{N} \langle\phi_1\phi_{i_2}|U|\phi_{j_1}\phi_{j_2}\rangle \, \text{per}(A;1,i_2|j_1,j_2) + \sum_{i_1\neq i_2,2}^{N}\sum_{j_2=2}^{N} \langle\phi_{i_1}\phi_{i_2}|U|\phi_1\phi_{j_2}\rangle \, \text{per}(A;i_1,i_2|1,j_2)$$

$$= \sum_{i_1\neq i_2,2}^{N}\sum_{j_1\neq j_2,2}^{N} \langle\phi_1\phi_{i_2}|U|\phi_{j_1}\phi_{j_2}\rangle\langle\phi_{i_1}|\phi_1\rangle \, \text{per}(A;1,i_1,i_2|1,j_1,j_2)$$

$$+ \sum_{i_1\neq i_2,2}^{N}\sum_{j_1\neq j_2,2}^{N} \langle\phi_1|\phi_{j_1}\rangle\langle\phi_{i_1}\phi_{i_2}|U|\phi_1\phi_{j_2}\rangle \, \text{per}(A;1,i_1,i_2|1,j_1,j_2). \tag{A.7}$$

In the third case, none $\phi_1$ is associated with $U$. We have

$$\frac{1}{2}\sum_{i_1\neq i_2,2}^{N}\sum_{j_1\neq j_2,2}^{N} \langle\phi_{i_1}\phi_{i_2}|U|\phi_{j_1}\phi_{j_2}\rangle \, \text{per}(A;i_1,i_2|j_1,j_2)$$

$$= \frac{1}{2}\sum_{i_1\neq i_2,2}^{N}\sum_{j_1\neq j_2,2}^{N} \langle\phi_{i_1}\phi_{i_2}|U|\phi_{j_1}\phi_{j_2}\rangle\langle\phi_1|\phi_1\rangle \, \text{per}(A;1,i_1,i_2|1,j_1,j_2)$$

$$+ \frac{1}{2}\sum_{i_1\neq i_2\neq i_3,2}^{N}\sum_{j_1\neq j_2\neq j_3,2}^{N} \langle\phi_{i_1}\phi_{i_2}|U|\phi_{j_1}\phi_{j_2}\rangle\langle\phi_1|\phi_{j_3}\rangle\langle\phi_{i_3}|\phi_1\rangle \, \text{per}(A;1,i_1,i_2,i_3|1,j_1,j_2,j_3). \tag{A.8}$$

In the last line, we see that we have to calculate the permanents of a sereis of $(N-4) \times (N-4)$ matrices $((N-1)(N-2)(N-3)/6)^2$ times. By the improved Ryser algorithm, the total evaluation is on the order of $N^7 2^{(N-5)}$. This is the most time-consuming part of the calculation.

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
