# Peer review of "Permanent variational wave functions for bosons"

_SciPost Physics_

## Round 1 · Referee Report · Anonymous · 2021-10-3

Strengths

1) The paper is well written and carefully laid out.
2) I think that the paper is quite useful for someone who would like to learn more about permanents for bosonic systems.

Weaknesses

1) The paper does not solve the exponential scaling of the evaluation of the permanent.
2) It is not clear what problem is being solved by the work in the paper.
3) Context is not provided; why this instead of Monte Carlo algorithms for example?

Report

The authors explore the use of permanents for bosonic problems. The paper is well written and carefully laid out. I think that the paper is quite useful for someone who would like to learn more about permanents for bosonic systems.

Looking at the criteria for acceptance, I have trouble finding which one of them would apply to this paper:

1. Detail a groundbreaking theoretical/experimental/computational discovery;

The paper does not solve the exponential scaling of the evaluation of the permanent.

2. Present a breakthrough on a previously-identified and long-standing research stumbling block;

They do not enable larger or more accurate solutions of bosonic systems. It's not clear to me what problem they are trying to solve. Is it just understanding how a permanent performs? Why is that an important problem if so? Perhaps, if this were explained, the paper would be more suitable.

3. Open a new pathway in an existing or a new research direction, with clear potential for multipronged follow-up work;

It's not clear to me why someone would use this algorithm over Monte Carlo.

4. Provide a novel and synergetic link between different research areas.

I'm not sure how this could apply.

The work is done very carefully and thoroughly, but from my perspective, I do not see what problem they are solving and what use this work will have to people in the future. Perhaps the authors can explain this better in a future version of the paper.

Some other comments follow:

I'm confused about the fact that they do not mention Monte Carlo methods. There is no sign problem in bosons, so thousands of particles have been simulated exactly in the literature.

They compare the overlap between the permanent and the true ground state. It's not clear to me what this means--the overlap goes to zero as the system

Similarly, the claim of agreement with total energy and the one-body properties of the exact wave function is missing some context. In fermionic systems, it is common for a Slater determinant to obtain 99% of the total energy and similar single-particle physics, but that is not considered a good wave function because it is missing a lot of important physics. Do the authors have a way of establishing a context in which a permanent is a useful approximation?

  • validity: high
  • significance: low
  • originality: ok
  • clarity: high
  • formatting: excellent
  • grammar: excellent

Author:  Jiang-min Zhang  on 2021-11-18  [id 1953]

(in reply to Report 1 on 2021-10-03)

The authors explore the use of permanents for bosonic problems. The paper is well written and carefully laid out. I think that the paper is quite useful for someone who would like to learn more about permanents for bosonic systems.

Looking at the criteria for acceptance, I have trouble finding which one of them would apply to this paper:

  1. Detail a groundbreaking theoretical/experimental/computational discovery;

The paper does not solve the exponential scaling of the evaluation of the permanent.

Our reply: Our work has nothing to do with the big phrase 'ground-breaking'. We just made some progress in algorithm, and got some small but surprising findings.

  1. Present a breakthrough on a previously-identified and long-standing research stumbling block;

They do not enable larger or more accurate solutions of bosonic systems. It's not clear to me what problem they are trying to solve. Is it just understanding how a permanent performs? Why is that an important problem if so? Perhaps, if this were explained, the paper would be more suitable.

Our reply: For fermions, these two problems should be deemed basic, (i) for a given system, which Slater determinant will minimize the energy? (ii) for a given wave function, which Slater determinant is the best approximation of it, i.e., having the largest overlap with it? For both problems, one is to find the minimum of a smooth function defined on a Grassmannian manifold. The first problem leads to the Hartree-Fock method for fermions. The second problem (motivated by the problem of the entanglement in a fermionic wave function) leads to the notion of the best Slater approximation and was considered in

https://journals.aps.org/prl/abstract/10.1103/PhysRevLett.109.186401

https://journals.aps.org/pra/abstract/10.1103/PhysRevA.89.012504

https://journals.aps.org/pra/abstract/10.1103/PhysRevA.94.032513

Recently, this notion finds use in the study of many-body localization, see

https://journals.aps.org/prb/abstract/10.1103/PhysRevB.104.205411

For bosons, we have two similar problems, (i) for a given system, which permanent state will minimize the energy? (ii) for a given wave function, which permanent state is the best approximation of it? The fundamental difference than the fermionic case is that now the objective functions are defined not on a manifold.

These are the two problems we are trying to solve. As mentioned in the Introduction, a few people considered the 1st problem before, but they either failed to find an algorithm, or failed to put the algorithm in a transparent and practical way. We think we achieved this. In particular, we are the first to be able to handle the multi-configuration case. In comparison with those people, we indeed can handle larger systems and get more accurate numbers.

An important objective of the paper is indeed understanding how a permanent performs. (i) Due to the symmetry constraint, there might be some hidden structure in a bosonic wave function. At least for fermions, this is the case. The anti-symmetry condition has consequences far beyond the naive Pauli exclusion principle. See

https://physics.aps.org/articles/v6/8

for a glimpse. Our idea is to study the structure of a bosonic state by studying its optimal permanent approximation. (ii) To get good numbers, we had better have a good understanding of the structure of the true ground state.

Finally, we just want to say that any optimization problem simply defined should be of importance. Good problems are simple problems. Good objects are simple objects. Permanent states are simple but not easy.

  1. Open a new pathway in an existing or a new research direction, with clear potential for multipronged follow-up work;

It's not clear to me why someone would use this algorithm over Monte Carlo.

Our Reply: The algorithm is meant for a few-body system, not a many-body system. Note that besides the many-body community, there is also a few-body community, in which devising algorithms to get numbers more and more accurately is an ever-lasting endeavor, see for example https://journals.aps.org/rmp/pdf/10.1103/RevModPhys.85.693. Some people spend their whole life studying two-particle or three-particle systems.

MC is superior for large N, but for small N, likely it is much slower than the current algorithm. For example, in Fig.24a, to solve the ground state energy of the two-particle system on a 15*15*15 lattice, on our laptop, it takes only 20 seconds, and the energy precision is 0.05%. We do not know how is it with MC. But the point is that by the algorithm, we got know that the true ground state can be exceedingly well approximated by a permanent state (overlap about 0.9995). This knowledge can hardly be gained with MC.

That the Monte Carlo can work does not mean it is efficient. Consider a single particle in a 1d trap. To get the ground state, a naive finite difference method (or a finite element method) is superior to a sophisticated Monte Carlo. Our algorithm is like such a direct method.

With the algorithm, we can numerically search for the optimal permanent approximation of a bosonic wave function, the overlap will provide a geometric measure of the entanglement in the wave function. Hence, it might be of interest for people in quantum information. MC cannot serve such purposes.

Anyway, it is unlikely that one approach is always better than another approach with completely different perspectives and methodology.

  1. Provide a novel and synergetic link between different research areas.

I'm not sure how this could apply.

The work is done very carefully and thoroughly, but from my perspective, I do not see what problem they are solving and what use this work will have to people in the future. Perhaps the authors can explain this better in a future version of the paper.

Our reply: We are concerned about the two basic problems mentioned above. The paper is actually a by-product of our investigation of the open problems listed in the conclusion part.

The algorithm might be of use for the following purposes

(i)For quantum information people, the algorithm can be used to find the best permanent approximation of a bosonic wave function, and thus determine the geometric entanglement in the wave function.

  1. The algorithm can also work for fermions. People doing Hartree-Fock for fermions often find that the iteration does not converge, as they update all the orbitals simultaneously each time. See

    https://physics.stackexchange.com/questions/20703/why-does-iteratively-solving-the-hartree-fock-equations-result-in-convergence

for a discussion. If instead a single orbital is updated each time, convergence can be guaranteed. Moreover, it is possible that a fermionic wave function can be put in the compact form of

Psi = Determinant_1 + Determinant_2 + Determinant_3

but with the single-particle orbitals used to construct the Slater determinants non-orthogonal. If one persists to use orthogonal single-particle orbitals, the list of Fock states should be much longer. For such states, the algorithm should work well. This is what we are going to do next.

  1. It should be possible to combine the algorithm with the Lanczos method. Usually, a Lanczos process starts with a random vector. Now we can think of starting from a permanent state fed by the algorithm, and at each step we generate new permanent states. We tested this idea for N=2. It seems to be promising.

  2. We really cherish the first open problem in the conclusion part. It motivated the whole paper. It is a simple, good, worthy, but not easy problem.

  3. We have seen that for many models, the ground state can be well approximated by permanent states. The next question is, how about a time-evolving state? If still so, permanent states can be used for dynamical problems.

Some other comments follow:

I'm confused about the fact that they do not mention Monte Carlo methods. There is no sign problem in bosons, so thousands of particles have been simulated exactly in the literature.

Our reply: We are sorry that we did not make it clear that we are targetting few-body systems. For such systems, we are confident that our method is faster than Monte Carlo and can provide more information. Anyway, we have a wave function in the end, while MC only delivers a few numbers.

By the way, MC does not necessarily work well for bosons. There can also be a sign problem in the presence of frustration. See e.g.,

https://arxiv.org/pdf/1912.12464.pdf

But this is never an issue for us. MC might have sign problem for fermions, but our algorithm apparently can be modified for fermions.

We have added a subsection (the one including equation 74), in which we study a two-particle system. The permanent approximation is amazingly good (overlap 0.9995). That is totally unexpected. We do not know whether the same finding can be gained with Monte Carlo. Monte Carlo can deliver numbers, but not necessarily the structure of the wave function.

They compare the overlap between the permanent and the true ground state. It's not clear to me what this means--the overlap goes to zero as the system

Our reply: Yes. As the particle number N increases to infinity, even the best variational wave function one can think of will have zero overlap with the exact ground state. But for a specific value of N, how large is it at all? This is a meaningful question for a few-body system. It quantifies the goodness of a variational wave function, and yields information of the structure of the true ground state.

Our finding is that even with 12 bosons and 12 sites, the overlap is still at least 0.96 regardless of the value of the onsite interaction strength g. This should be impressive, right?Note that the dim of the Hilbert space is as large as 1.35 million. For 20 bosons and 20 sites, we cannot do exact diagonalization to get the exact ground state, so we cannot calculate the overlap, but by extrapolation, one can infer that it is at least 0.9. This is really a big number for a not so small system. The Hilbert space is now 6.9*10^10. It takes huge memory to store the exact wave function, but minimal memeory to store its permanent approximation. Very large data compression ratio yet very high fidelity. We are frustrated that some people seem not be impressed. The only other case of high overlap between the variational state and the exact state is the Laughlin wave function, as far as we know.

We calculate these numbers to demonstrate that the permanent is really a good approximation of the exact ground state, good even on the wave function level, not just on the energy level.

We stress that we have a wave function based method. We have an explicit variational wave function. Therefore, we can calculate the overlap. That is an advantage over many non-wave function based method, like many Monte Carlo methods. Those methods might also deliver the ground state energy well, but get the wave function poor, if they can provide a wave function at all.

Similarly, the claim of agreement with total energy and the one-body properties of the exact wave function is missing some context. In fermionic systems, it is common for a Slater determinant to obtain 99% of the total energy and similar single-particle physics, but that is not considered a good wave function because it is missing a lot of important physics. Do the authors have a way of establishing a context in which a permanent is a useful approximation?

Our reply: No, we never take the agreement in total energy as a big deal. Our emphasis is always on the overlap and the one-body correlation function (reduced density matrix). Anyway, even if the variational wave function is the first excited state and thus orthogonal to the true ground state, it can get the energy very accurate.

Hence, the overlap is a much more stringent test of the closeness between the variaitonal wave function and the true ground state than energy, and it has always been our emphasis. In most cases in the paper, the overlap is at least 0.96, and in some cases, it could even be 0.9996. We think these numbers are sufficient to prove that a permanent is a useful approximation.

According to our impression, the 99% precision in energy generally cannot be achieved with a single Slater determinant. It is more likely 90%. For other quantities, the precision is even worse (error in energy is 2nd order, error in other quantities 1st order). We tried to find the overlap in the Hartree-Fock literature, but failed.

We did not state it in the paper (to be safe, we need more cases), but it is our feeling that, permanents work for bosons better than determinants work for fermions. The reason might be that, a bosonic ground state is positive everywhere and so there is no cancellation in the overlap integral.

System Message: ERROR/3 (<string>, line 141)

Document may not end with a transition.

---

## Round 1 · Referee Report · Anonymous · 2021-10-10

Strengths

1. Accurate presentation of the permanent and of its properties.
2. Clear and well-written presentation.

Weaknesses

1. Lacking a comparison with alternative numerical methods.
2. Limited to small systems, without a clear statement of what is expected for larger lattices.
3. Not specifying what is the main point of relevance of this reserach.

Report

In this paper the authors discuss the construction of variational states for bosonic systems as of permanent states (that is, the symmetric analog, for bosonic systems, of the Slater determinant for fermionic systems). Specifically, after a main presentation of their formalism, the authors apply their framework to the specific example of the one-dimensional Bose-Hubbard model, both with periodic and open boundary conditions. The specific quantities they are interested in are the variational energy, that is, the average value of the system Hamiltonian over the variational state (a quantity that has to be minimized to optimize the procedure) and, when available, the overlap integral with the true groundstate of the system (apparently, a quantity that has to be the closest is possible to the ideal value of 1).

A first case study considered is the one-dimensional Bose-Hubbard model with filling one, over a lattice with L=12 sites. According to what the authors state, the results appear to be encouraging, both for what concerns the energy estimate, as well as the overlap with the exact wavefunction. In particular, as they use a permanent state built out of exponential, as well as Lorenzian, wavefunctions, they find that the Lorenzian/exponential wavefunctions work better than the exponential/Lorenzian wavefunctions, at small/large values of the interaction parameter g. Having stated their results for a well-defined choice of the single particle orbitales determining the permanent state, the authors then move to the more general situation of not starting with a preassigned set of orbitals, but rather determining them by following the optimization algorithm they discuss in section IV. In that section, the authors discuss various issues related to the convergence of the whole numerical procedure concluding that, for instance, the convergence is slower the weaker the interaction and the larger the number of configurations. Eventually, in section V, a number of applications of the whole procedure is presented to the Bose-Hubbard model with both periodic and open boundary conditions, as well as to the same model with an additional harmonic confinement potential, always finding a good consistency with the exact results derived within the same models.

Within the concluding remarks, the authors provide a discussion about the reliability of their technique, especially in view of the quest of optimizing the numerical effort in deriving a good approximation to the true groundstate of the system.

Coming to my judgement about this manuscript, in my opinion, as it stands now, it provides no more than an accurate, and well written, presentation of the properties of the permanent and of its implementation for numerical calculations. Aside for that, I can hardly see why this paper should warrant publication in SciPost. To motivate my opinion, let me list some specific remarks I have about this manuscript, in relation to the SciPost expectations and main acceptance criteria:

Expectations:

1. Detail a groundbreaking theoretical/experimental/computational discovery:

The paper clearly deaks with a computation method. In order to see a specific advancement on the computational side I think a careful comparison should be made with alternative numerical methods for what concerns the typical system size one can deal with (of course for similar systems), the over-all time required to run algorithms computing the same thing in the same system with different techniques (such as, e.g., the authors’ method, the DMRG approach, the Monte Carlo techniques, and so on) to assess whether the method proposed here has, at least, the same level of reliability of alternative, well-grounded and widely used approaches. In this manuscript I see nothing pointing in that direction.

2. Present a breakthrough on a previously-identified and long-standing research stumbling block:

The proposed applications of the authors’ method are limited to a small number of well-known one-dimensional lattice models with a small number of sites (a lattice as large as about 10 sites) and even in that case the reliability of the results is apparently not better that what one would possibly get by using alternative numerical methods. I do not see any specific long-standing problem that might be (even partially) tackled by resorting to the technique proposed in this paper, better than with some alternative method.

3. Open a new pathway in an existing or a new research direction, with clear potential for multipronged follow-up work:

The only immediate possible follow-up work, along the direction of this manuscript, that I can see could be extending the system size and possibly checking the scaling of e.g. computation time, estimates of the energy and/or of the state overlap, and so on, with the system size. But, again, without a serious comparation to what one could do with alternative numerical methods I do not see why one should push forward this research line.

4. Provide a novel and synergetic link between different research areas:

I cannot see immediate chances for synergetic links between different areas arising from this work.

General acceptance criteria:

1. Be written in a clear and intelligible way, free of unnecessary jargon, ambiguities and misrepresentations:

The paper does meet all the above requirements: it is well written, free of unnecessary jargon and more-or-less self-contained, for what concerns the definition and the properties of the permanent.

2. Contain a detailed abstract and introduction explaining the context of the problem and objectively summarizing the achievements:

The context of the problem is explained at a good enough level of details. Instead, I do not believe that the its possible main achiements are clearly outlined (see previous section of my report).

3. Provide sufficient details (inside the bulk sections or in appendices) so that arguments and derivations can be reproduced by qualified experts:

The various mathematical steps are well documented and detailed. The paper does meet this requirement.

4. Provide citations to relevant literature in a way that is as representative and complete as possible:

The paper meets this requirement.

5. Provide (directly in appendices, or via links to external repositories) all reproducibility-enabling resources: explicit details of experimental protocols, datasets and processing methods, processed data and code snippets used to produce figures, etc.:

The paper meets this requirement.

6. Contain a clear conclusion summarizing the results (with objective statements on their reach and limitations) and offering perspectives for future work:

The paper meets this requirement, although the stated results and the proposed perspectives are pretty questionable (see previous sections of the report).

Requested changes

1. A careful comparison should be made with alternative numerical methods for what concerns the typical system size one can deal with.

2. The size of the physical systems studied here should at least be large enough to clearly state scaling properties with the system size.

3. An (even qualitative) discussion of whether the method is expected to be reliable when applied to higher-dimensional systems should be done.

  • validity: ok
  • significance: low
  • originality: low
  • clarity: good
  • formatting: good
  • grammar: good

Author:  Jiang-min Zhang  on 2021-11-18  [id 1952]

(in reply to Report 2 on 2021-10-10)

Strengths 1. Accurate presentation of the permanent and of its properties. 2. Clear and well-written presentation.

Weaknesses 1. Lacking a comparison with alternative numerical methods. 2. Limited to small systems, without a clear statement of what is expected for larger lattices. 3. Not specifying what is the main point of relevance of this reserach.

Report In this paper the authors discuss the construction of variational states for bosonic systems as of permanent states (that is, the symmetric analog, for bosonic systems, of the Slater determinant for fermionic systems). Specifically, after a main presentation of their formalism, the authors apply their framework to the specific example of the one-dimensional Bose-Hubbard model, both with periodic and open boundary conditions. The specific quantities they are interested in are the variational energy, that is, the average value of the system Hamiltonian over the variational state (a quantity that has to be minimized to optimize the procedure) and, when available, the overlap integral with the true groundstate of the system (apparently, a quantity that has to be the closest is possible to the ideal value of 1).

A first case study considered is the one-dimensional Bose-Hubbard model with filling one, over a lattice with L=12 sites. According to what the authors state, the results appear to be encouraging, both for what concerns the energy estimate, as well as the overlap with the exact wavefunction. In particular, as they use a permanent state built out of exponential, as well as Lorenzian, wavefunctions, they find that the Lorenzian/exponential wavefunctions work better than the exponential/Lorenzian wavefunctions, at small/large values of the interaction parameter g. Having stated their results for a well-defined choice of the single particle orbitales determining the permanent state, the authors then move to the more general situation of not starting with a preassigned set of orbitals, but rather determining them by following the optimization algorithm they discuss in section IV. In that section, the authors discuss various issues related to the convergence of the whole numerical procedure concluding that, for instance, the convergence is slower the weaker the interaction and the larger the number of configurations. Eventually, in section V, a number of applications of the whole procedure is presented to the Bose-Hubbard model with both periodic and open boundary conditions, as well as to the same model with an additional harmonic confinement potential, always finding a good consistency with the exact results derived within the same models.

Within the concluding remarks, the authors provide a discussion about the reliability of their technique, especially in view of the quest of optimizing the numerical effort in deriving a good approximation to the true ground state of the system.

Coming to my judgement about this manuscript, in my opinion, as it stands now, it provides no more than an accurate, and well written, presentation of the properties of the permanent and of its implementation for numerical calculations. Aside for that, I can hardly see why this paper should warrant publication in SciPost. To motivate my opinion, let me list some specific remarks I have about this manuscript, in relation to the SciPost expectations and main acceptance criteria:

Expectations:

  1. Detail a groundbreaking theoretical/experimental/computational discovery:

The paper clearly deals with a computation method. In order to see a specific advancement on the computational side I think a careful comparison should be made with alternative numerical methods for what concerns the typical system size one can deal with (of course for similar systems), the over-all time required to run algorithms computing the same thing in the same system with different techniques (such as, e.g., the authors' method, the DMRG approach, the Monte Carlo techniques, and so on) to assess whether the method proposed here has, at least, the same level of reliability of alternative, well-grounded and widely used approaches. In this manuscript I see nothing pointing in that direction.

Our reply: Sorry, but we cannot make such a detailed comparison for the following reasons:

  1. We are not experts in Monte Carlo or DMRG. We really cannot learn them from scratch now.

  2. It is apparent that the current algorithm has its advantages. DMRG works well only for 1d systems with short range hoppings and interactions. The current method works well for 3D, and the hopping and interaction can be arbitrary. As for MC, it can suffer from the negative sign problem if there is frustration, but that is not a problem for us. In many cases, it should be slower than us. Furthermore, the current algorithm can be easily modified for fermions, for which MC might meet great difficulty.

  3. It is apparent from the algorithm that the memory cost is minimal. It does not increases with the particle number N (unlike exact diagonalization). In a newly added subsection, we treat a two-particle system with L = 9261 sites. The memory cost is that of a 9261*9261 matrix. It is the same for N = 10.

  4. As for time, it can be seen that if the particle number N is around 10, the permanent computation is something. But it is still okay---in our paper, figure 12 (N = 11) is most time-consuming, but it is still less than 24 hours (it would be less than 8 hours if we just want fig. 12a). Most figures can be done in 30 mins with Matlab. For fixed N, if the system size L increases, solving the generalized eigenvalue problem could become the bottle neck in the end. But the complexity of solving a generalized eigenvalue problem increases as L^3 only. We have actually collected the relevant data, but we really do not want to display it because it can be inferred from the algorithm. We have actually discussed it in the conclusion part before.

  5. The current algorithm is a wave-function based method. Once we have the wave function, we can calculate any quantity we like. The philosophy is totally different from those of MC or DMRG. It is unlikely that a method can be substituted by another one with a completely different philosophy.

  1. Present a breakthrough on a previously-identified and long-standing research stumbling block:

The proposed applications of the authors' method are limited to a small number of well-known one-dimensional lattice models with a small number of sites (a lattice as large as about 10 sites) and even in that case the reliability of the results is apparently not better that what one would possibly get by using alternative numerical methods. I do not see any specific long-standing problem that might be (even partially) tackled by resorting to the technique proposed in this paper, better than with some alternative method.

Our reply: To prove that the algorithm can work for higher dimensions, we have added a subsection (the one containing equation 74). It is a cubic lattice, as large as 21*21*21. The permanent approximation is amazingly good. The algorithm is faster than exact diagonalization (ED). It can also handle bigger lattices than ED. With the algorithm, we got know that the exact ground state can be exceedingly well approximated with a single permanent. We do not know whether MC can deliver the same insight. Moreover, once we have a simple wave function, we can calculate any physical quantity as we like, there is no need to restart the algorithm.

In this tentative work, we first need to benchmark the algorithm against ED. We can easily handle N=12 particles on a lattice of size L = 100. It is as easy as the N=12 and L = 12 case. But for such a system size, we have no ED for comparison. Note that our primary concern is whether a ground state can be well approximated with permanent states---We really care about the overlap. So, we need the exact ground state, which can only be delivered by ED.

  1. Open a new pathway in an existing or a new research direction, with clear potential for multipronged follow-up work:

The only immediate possible follow-up work, along the direction of this manuscript, that I can see could be extending the system size and possibly checking the scaling of e.g. computation time, estimates of the energy and/or of the state overlap, and so on, with the system size. But, again, without a serious comparation to what one could do with alternative numerical methods I do not see why one should push forward this research line.

Our reply: To be frank, we are not interested in a detailed but laborious comparison. The current algorithm is in spirit like a Hartree-Fock method. The Hartree-Fock method is not replaced by other methods. The paper is already too long. Making it longer would only reduce the readability.

We can upload the codes to Github, so people can compare if they like.

As for possible follow-up work, please see our reply to the first referee.

  1. Provide a novel and synergetic link between different research areas:

I cannot see immediate chances for synergetic links between different areas arising from this work.

Our reply: Please see our reply to the first referee.

General acceptance criteria:

  1. Be written in a clear and intelligible way, free of unnecessary jargon, ambiguities and misrepresentations:

The paper does meet all the above requirements: it is well written, free of unnecessary jargon and more-or-less self-contained, for what concerns the definition and the properties of the permanent.

  1. Contain a detailed abstract and introduction explaining the context of the problem and objectively summarizing the achievements:

The context of the problem is explained at a good enough level of details. Instead, I do not believe that the its possible main achievements are clearly outlined (see previous section of my report).

Our reply: Sorry that we failed to express ourselves better. We have rewritten many paragraphs. See the list of changes.

  1. Provide sufficient details (inside the bulk sections or in appendices) so that arguments and derivations can be reproduced by qualified experts:

The various mathematical steps are well documented and detailed. The paper does meet this requirement.

  1. Provide citations to relevant literature in a way that is as representative and complete as possible:

The paper meets this requirement.

  1. Provide (directly in appendices, or via links to external repositories) all reproducibility-enabling resources: explicit details of experimental protocols, datasets and processing methods, processed data and code snippets used to produce figures, etc.:

The paper meets this requirement.

  1. Contain a clear conclusion summarizing the results (with objective statements on their reach and limitations) and offering perspectives for future work:

The paper meets this requirement, although the stated results and the proposed perspectives are pretty questionable (see previous sections of the report).

Our reply: Possibly the referee is skeptical about the applicability of the algorithm to higher dimensions? For sure it works perfectly for higher dimensions. We have added a subsection to demonstrate this point. See the subsection containing equation 74.

Requested changes 1. A careful comparison should be made with alternative numerical methods for what concerns the typical system size one can deal with.

  1. The size of the physical systems studied here should at least be large enough to clearly state scaling properties with the system size.

  2. An (even qualitative) discussion of whether the method is expected to be reliable when applied to higher-dimensional systems should be done.

Our reply: We hope this would not offend the referee. But we really do not think the required comparison necessary in this paper (but it is worthy in later work). Just look at the algorithm, it reduces a few-body problem to a single-particle problem. Therefore, the system size L is limited just by how much memory you have to store an L*L matrix and how much time your cpu needs to solve a generalized eigenvalue problem of size L.

We are grateful to the referee for the 3rd point though. While it is apparent that the algorithm works for arbitrary dimensions, we are indeed motivated to study a meaningful 3d problem---a stability problem. See the subsection containing equation 74. We were shocked by the numbers.

System Message: ERROR/3 (<string>, line 134)

Document may not end with a transition.

Author:  Jiang-min Zhang  on 2021-11-18  [id 1951]

(in reply to Report 2 on 2021-10-10)
Category:
remark
answer to question

The authors explore the use of permanents for bosonic problems. The paper is well written and carefully laid out. I think that the paper is quite useful for someone who would like to learn more about permanents for bosonic systems.

Looking at the criteria for acceptance, I have trouble finding which one of them would apply to this paper:

  1. Detail a groundbreaking theoretical/experimental/computational discovery;

The paper does not solve the exponential scaling of the evaluation of the permanent.

Our reply: Our work has nothing to do with the big phrase "ground-breaking". We just made some progress in algorithm, and got some small but surprising findings.

  1. Present a breakthrough on a previously-identified and long-standing research stumbling block;

They do not enable larger or more accurate solutions of bosonic systems. It's not clear to me what problem they are trying to solve. Is it just understanding how a permanent performs? Why is that an important problem if so? Perhaps, if this were explained, the paper would be more suitable.

Our reply: For fermions, these two problems should be deemed basic, (i) for a given system, which Slater determinant will minimize the energy? (ii) for a given wave function, which Slater determinant is the best approximation of it, i.e., having the largest overlap with it? For both problems, one is to find the minimum of a smooth function defined on a Grassmannian manifold. The first problem leads to the Hartree-Fock method for fermions. The second problem (motivated by the problem of the entanglement in a fermionic wave function) leads to the notion of the best Slater approximation and was considered in

https://journals.aps.org/prl/abstract/10.1103/PhysRevLett.109.186401

https://journals.aps.org/pra/abstract/10.1103/PhysRevA.89.012504

https://journals.aps.org/pra/abstract/10.1103/PhysRevA.94.032513

Recently, this notion finds use in the study of many-body localization, see

https://journals.aps.org/prb/abstract/10.1103/PhysRevB.104.205411

For bosons, we have two similar problems, (i) for a given system, which permanent state will minimize the energy? (ii) for a given wave function, which permanent state is the best approximation of it? The fundamental difference than the fermionic case is that now the objective functions are defined not on a manifold.

These are the two problems we are trying to solve. As mentioned in the Introduction, a few people considered the 1st problem before, but they either failed to find an algorithm, or failed to put the algorithm in a transparent and practical way. We think we achieved this. In particular, we are the first to be able to handle the multi-configuration case. In comparison with those people, we indeed can handle larger systems and get more accurate numbers.

An important objective of the paper is indeed understanding how a permanent performs. (i) Due to the symmetry constraint, there might be some hidden structure in a bosonic wave function. At least for fermions, this is the case. The anti-symmetry condition has consequences far beyond the naive Pauli exclusion principle. See

https://physics.aps.org/articles/v6/8

for a glimpse. Our idea is to study the structure of a bosonic state by studying its optimal permanent approximation. (ii) To get good numbers, we had better have a good understanding of the structure of the true ground state.

Finally, we just want to say that any optimization problem simply defined should be of importance. Good problems are simple problems. Good objects are simple objects. Permanent states are simple but not easy.

  1. Open a new pathway in an existing or a new research direction, with clear potential for multipronged follow-up work;

It's not clear to me why someone would use this algorithm over Monte Carlo.

Our Reply: The algorithm is meant for a few-body system, not a many-body system. Note that besides the many-body community, there is also a few-body community, in which devising algorithms to get numbers more and more accurately is an ever-lasting endeavor, see for example https://journals.aps.org/rmp/pdf/10.1103/RevModPhys.85.693. Some people spend their whole life studying two-particle or three-particle systems.

MC is superior for large N, but for small N, likely it is much slower than the current algorithm. For example, in Fig.24a, to solve the ground state energy of the two-particle system on a 15*15*15 lattice, on our laptop, it takes only 20 seconds, and the energy precision is 0.05%. We do not know how is it with MC. But the point is that by the algorithm, we got know that the true ground state can be exceedingly well approximated by a permanent state (overlap about 0.9995). This knowledge can hardly be gained with MC.

That the Monte Carlo can work does not mean it is efficient. Consider a single particle in a 1d trap. To get the ground state, a naive finite difference method (or a finite element method) is superior to a sophisticated Monte Carlo. Our algorithm is like such a direct method.

With the algorithm, we can numerically search for the optimal permanent approximation of a bosonic wave function, the overlap will provide a geometric measure of the entanglement in the wave function. Hence, it might be of interest for people in quantum information. MC cannot serve such purposes.

Anyway, it is unlikely that one approach is always better than another approach with completely different perspectives and methodology.

  1. Provide a novel and synergetic link between different research areas.

I'm not sure how this could apply.

The work is done very carefully and thoroughly, but from my perspective, I do not see what problem they are solving and what use this work will have to people in the future. Perhaps the authors can explain this better in a future version of the paper.

Our reply: We are concerned about the two basic problems mentioned above. The paper is actually a by-product of our investigation of the open problems listed in the conclusion part.

The algorithm might be of use for the following purposes

(i)For quantum information people, the algorithm can be used to find the best permanent approximation of a bosonic wave function, and thus determine the geometric entanglement in the wave function.

  1. The algorithm can also work for fermions. People doing Hartree-Fock for fermions often find that the iteration does not converge, as they update all the orbitals simultaneously each time. See

    https://physics.stackexchange.com/questions/20703/why-does-iteratively-solving-the-hartree-fock-equations-result-in-convergence

for a discussion. If instead a single orbital is updated each time, convergence can be guaranteed. Moreover, it is possible that a fermionic wave function can be put in the compact form of

Psi = Determinant_1 + Determinant_2 + Determinant_3

but with the single-particle orbitals used to construct the Slater determinants non-orthogonal. If one persists to use orthogonal single-particle orbitals, the list of Fock states should be much longer. For such states, the algorithm should work well. This is what we are going to do next.

  1. It should be possible to combine the algorithm with the Lanczos method. Usually, a Lanczos process starts with a random vector. Now we can think of starting from a permanent state fed by the algorithm, and at each step we generate new permanent states. We tested this idea for N=2. It seems to be promising.

  2. We really cherish the first open problem in the conclusion part. It motivated the whole paper. It is a simple, good, worthy, but not easy problem.

(v) We have seen that for many models, the ground state can be well approximated by permanent states. The next question is, how about a time-evolving state? If still so, permanent states can be used for dynamical problems.

System Message: WARNING/2 (<string>, line 96)

Title underline too short.

(v) We have seen that for many models, the ground state can be well approximated by permanent states. The next question is, how about a time-evolving state? If still so, permanent states can be used for dynamical problems.
%%%%%%%%%%%%%%%%%%

Some other comments follow:

I'm confused about the fact that they do not mention Monte Carlo methods. There is no sign problem in bosons, so thousands of particles have been simulated exactly in the literature.

Our reply: We are sorry that we did not make it clear that we are targetting few-body systems. For such systems, we are confident that our method is faster than Monte Carlo and can provide more information. Anyway, we have a wave function in the end, while MC only delivers a few numbers.

By the way, MC does not necessarily work well for bosons. There can also be a sign problem in the presence of frustration. See e.g.,

https://arxiv.org/pdf/1912.12464.pdf

But this is never an issue for us. MC might have sign problem for fermions, but our algorithm apparently can be modified for fermions.

We have added a subsection (the one including equation 74), in which we study a two-particle system. The permanent approximation is amazingly good (overlap 0.9995). That is totally unexpected. We do not know whether the same finding can be gained with Monte Carlo. Monte Carlo can deliver numbers, but not necessarily the structure of the wave function.

They compare the overlap between the permanent and the true ground state. It's not clear to me what this means--the overlap goes to zero as the system

Our reply: Yes. As the particle number N increases to infinity, even the best variational wave function one can think of will have zero overlap with the exact ground state. But for a specific value of N, how large is it at all? This is a meaningful question for a few-body system. It quantifies the goodness of a variational wave function, and yields information of the structure of the true ground state.

Our finding is that even with 12 bosons and 12 sites, the overlap is still at least 0.96 regardless of the value of the onsite interaction strength g. This should be impressive, right?Note that the dim of the Hilbert space is as large as 1.35 million. For 20 bosons and 20 sites, we cannot do exact diagonalization to get the exact ground state, so we cannot calculate the overlap, but by extrapolation, one can infer that it is at least 0.9. This is really a big number for a not so small system. The Hilbert space is now 6.9*10^10. It takes huge memory to store the exact wave function, but minimal memeory to store its permanent approximation. Very large data compression ratio yet very high fidelity. We are frustrated that some people seem not be impressed. The only other case of high overlap between the variational state and the exact state is the Laughlin wave function, as far as we know.

We calculate these numbers to demonstrate that the permanent is really a good approximation of the exact ground state, good even on the wave function level, not just on the energy level.

We stress that we have a wave function based method. We have an explicit variational wave function. Therefore, we can calculate the overlap. That is an advantage over many non-wave function based method, like many Monte Carlo methods. Those methods might also deliver the ground state energy well, but get the wave function poor, if they can provide a wave function at all.

Similarly, the claim of agreement with total energy and the one-body properties of the exact wave function is missing some context. In fermionic systems, it is common for a Slater determinant to obtain 99% of the total energy and similar single-particle physics, but that is not considered a good wave function because it is missing a lot of important physics. Do the authors have a way of establishing a context in which a permanent is a useful approximation?

Our reply: No, we never take the agreement in total energy as a big deal. Our emphasis is always on the overlap and the one-body correlation function (reduced density matrix). Anyway, even if the variational wave function is the first excited state and thus orthogonal to the true ground state, it can get the energy very accurate.

Hence, the overlap is a much more stringent test of the closeness between the variaitonal wave function and the true ground state than energy, and it has always been our emphasis. In most cases in the paper, the overlap is at least 0.96, and in some cases, it could even be 0.9996. We think these numbers are sufficient to prove that a permanent is a useful approximation.

According to our impression, the 99% precision in energy generally cannot be achieved with a single Slater determinant. It is more likely 90%. For other quantities, the precision is even worse (error in energy is 2nd order, error in other quantities 1st order). We tried to find the overlap in the Hartree-Fock literature, but failed.

We did not state it in the paper (to be safe, we need more cases), but it is our feeling that, permanents work for bosons better than determinants work for fermions. The reason might be that, a bosonic ground state is positive everywhere and so there is no cancellation in the overlap integral.

System Message: ERROR/3 (<string>, line 142)

Document may not end with a transition.

---

## Editorial Decision

resubmitted